# EP-ADTA: Edge Prediction-Based Adaptive Data Transfer Algorithm for Underwater Wireless Sensor Networks (UWSNs)

**DOI:** 10.3390/s22155490

**Published:** 2022-07-23

**Authors:** Bin Wang, Kerong Ben, Haitao Lin, Mingjiu Zuo, Fengchen Zhang

**Affiliations:** College of Electronic Engineering, Naval University of Engineering, Wuhan 430033, China; benkerong08@163.com (K.B.); figue2015@163.com (H.L.); zuomingjiu@126.com (M.Z.); fczhang0606@163.com (F.Z.)

**Keywords:** underwater wireless sensor networks (UWSNs), underwater monitoring applications, edge computing, intelligent routing algorithm, reinforcement learning (RL), auto regressive and moving average (ARMA)

## Abstract

The underwater wireless sensor network is an important component of the underwater three-dimensional monitoring system. Due to the high bit error rate, high delay, low bandwidth, limited energy, and high dynamic of underwater networks, it is very difficult to realize efficient and reliable data transmission. Therefore, this paper posits that it is not enough to design the routing algorithm only from the perspective of the transmission environment; the comprehensive design of the data transmission algorithm should also be combined with the application. An edge prediction-based adaptive data transmission algorithm (EP-ADTA) is proposed that can dynamically adapt to the needs of underwater monitoring applications and the changes in the transmission environment. EP-ADTA uses the end–edge–cloud architecture to define the underwater wireless sensor networks. The algorithm uses communication nodes as the agents, realizes the monitoring data prediction and compression according to the edge prediction, dynamically selects the transmission route, and controls the data transmission accuracy based on reinforcement learning. The simulation results show that EP-ADTA can meet the accuracy requirements of underwater monitoring applications, dynamically adapt to the changes in the transmission environment, and ensure efficient and reliable data transmission in underwater wireless sensor networks.

## 1. Introduction

Underwater wireless sensor networks are widely used in early disaster warning, pollutant monitoring, hydrological data monitoring and collection, marine resource exploration, auxiliary navigation, and marine military applications [1], and have become an important infrastructure component of the underwater Internet of Things [2]. Underwater wireless sensor networks consist of two parts, a surface wireless sensor network and an underwater wireless sensor network, and include three-dimensional (3D) mobile, fixed, or mixed networks. Data are collected by underwater sensor nodes, which can be transmitted from underwater to surface sink nodes in the form of hop-by-hop forwarding through communication nodes at different depths, or collected by AUV and then sent to the surface. The surface network part uses a radio to realize the information transmission between the underwater network and the shore-based data center. Although underwater data can be transmitted by radio, wireless light, sound, and other media, underwater long-distance data transmission still relies on the underwater acoustic mode.

Compared with terrestrial wireless sensor networks based on RF communication, underwater wireless sensor networks are characterized by an unstable, uneven transmission environment; many interference factors; a small transmission channel capacity; a large delay; low reliability; limited energy of communication nodes; a limited energy supply; and a high degree of space–time dynamics [3,4]; as a result of these characteristics, it is very difficult to realize underwater wireless network communication. At present, the underwater wireless channel is considered to be one of the most difficult-to-manage channels. In order to overcome the adverse effects of the underwater transmission environment, researchers have divided the protocol stack of underwater wireless sensor networks into a physical layer, data link layer, network layer, transmission layer, and application layer, and carried out cross-layer optimization design. They have adopted the methods of channel awareness [5,6], energy consumption awareness [7,8,9,10], and delay awareness [5,11] to better select transmission channels. Layered [9,12] or clustering [12,13,14,15] methods have been adopted to divide the underwater network, balance the energy consumption, and extend the network life. Previous studies have endowed the transmission routing with intelligence, so that it can adapt to the changes in the transmission environment; optimize the transmission quality; bypass network holes [16,17]; and realize efficient, reliable, and stable data transmission. However, due to the complex, dynamic, and extremely limited underwater transmission environment, when devising underwater monitoring applications having a stable and huge data transmission demand, it is not enough to only pay attention to the design and combination optimization of the physical layer, link layer, and network layer of the routing protocol. The uncontrolled demand for transmission of a large quantity of high-precision monitoring data will make the routing protocols ineffective due to congestion and energy depletion. The cross-layer design of routing protocols, which is not closely combined with the application layer, cannot maximize the use of transmission resources and achieve effective data transmission [18,19]. It is necessary to design an adaptive data transmission method that can dynamically adjust the accuracy of data transmission according to the transmission conditions and application requirements while correctly selecting the relay node.

End–edge–cloud architecture [20,21], which is a hierarchical architecture based on terminal equipment, the edge network, and the cloud environment, has been gradually applied to the Internet of Things system. By replacing centralized cloud computing with edge computing dispersed in the edge network, this approach can better meet the needs of delay-sensitive applications, reduce data transmission load, and improve the overall performance of the network [22]. The underwater wireless sensor network is also applicable to the end–edge–cloud architecture. As a result of the improvement in the computing power of underwater nodes, edge computing will play a greater role in improving the performance of underwater wireless sensor networks [23,24]. Edge computing has been applied to underwater mobile wireless sensor networks composed of AUV and sensor nodes to reduce the data transmission burden between sensor nodes and the AUV, and between the AUV and shore-based data centers [24,25]. In the fixed network, the sensor nodes used for data collection can be defined as the information end; the wireless communication network is composed of underwater sensor nodes; communication nodes and sink nodes form the edge network; the shore-based data center network comprises the cloud environment; and the sink node is used as the gateway node connecting the underwater edge network and the cloud environment. During the process of underwater monitoring, the data are collected by the underwater sensor nodes, forwarded to the surface sink nodes via each hop through the underwater communication nodes, and then transmitted to the shore-based data center via satellite. The continuous data transmission demand for underwater monitoring applications has led to a more serious burden on the previously limited underwater wireless communication. However, as time series data and monitoring data generally show a great correlation and redundancy in time and space [26,27], the new monitoring data can be predicted in the edge network using an edge prediction algorithm and historical data. After edge prediction, only the prediction parameters and correction data that meet the application accuracy requirements should be transmitted, which can effectively compress the actual data transmission volume, thereby reducing the transmission delay and energy consumption, improving the packet-sending success rate, and enhancing the communication performance of underwater wireless sensor networks.

Agent modeling is a method used to divide and conquer complex systems. Intelligent agents can respond and process in a collaborative manner based on the intelligent policies by understanding the environment and other agent statuses [28,29]. The communication of the underwater wireless sensor network is complex and difficult. Each node in the edge network can be defined as an intelligent agent [30]. Through cooperation and communication, intelligent agents can effectively improve the performance of underwater wireless network communication [31,32]. Mobile agents have been used to enhance the network scalability [12]. Multiple agents have been used to optimize routing selection [28,29] and to improve the rationality of resource allocation [30,31]. Reinforcement learning is an important method for improving agent intelligence. It allows agents to learn the optimal strategy through trial-and-error training. As a result of the changes in the transmission channel, transmission node performance, and application transmission requirements, the agent must dynamically adjust the routing strategy and accuracy of data transmission according to the environmental state to ensure the performance of underwater communication. Over a certain period, reinforcement learning can equip agents with this ability [28,29]. Therefore, the data transmission strategy is based on reinforcement learning, which incorporates the combination of the environmental status and business requirements into the state space, and selects the best data transmission accuracy while selecting the best relay node, to obtain a data transmission strategy. Under the requirements of underwater monitoring data accuracy, the number of transmitted correction data will be dynamically adjusted according to the change in the transmission environment. When the transmission environment is good, more correction data will be transmitted to ensure higher transmission accuracy. When the transmission environment is poor, the number of transmitted correction data can also be dynamically reduced to ensure a higher packet-transmission success rate. When selecting the best relay node, the data transmission strategy comprehensively considers the location of neighbor nodes, residual energy, and transmission channel delay. In order to avoid too many nodes participating in reinforcement learning and improve the convergence speed of reinforcement learning, a virtual pipeline is built between the source node and the target node based on the residual energy, and the relay node is only selected from the virtual pipeline. The environment and neighbor agent statuses are updated by the monitor and control channel established based on the broadcast between neighbor nodes. The agent based on reinforcement learning can independently find the best data transmission strategy according to the change in the environmental state, and realize efficient and reliable underwater wireless network communication.

In the end–edge–cloud and agent modeling architecture, with the comprehensive design of all function layers of underwater network, this paper presents an adaptive data transmission algorithm (EP-ADTA) based on edge prediction and reinforcement learning. The main contributions include the following:

(1) The end–edge–cloud architecture is used to define the fixed three-dimensional underwater wireless sensor network, and edge computing is applied to the data transmission of the underwater wireless sensor network. A new method of realizing the efficient data transmission of the underwater wireless sensor network by replacing communication with computing is proposed. 

(2) Taking underwater time series data monitoring as an example, a set of edge predictions based on the ARMA model is proposed to predict the new monitoring data at the edge network. Edge prediction cannot only automatically match the optimal prediction parameters, but also dynamically adjust the amount of correction data to be transmitted according to the accuracy changes in the application data.

(3) Based on agent modeling technology, a data transmission algorithm based on reinforcement learning is proposed. The intelligent agent located in the edge network can sense the changes in the transmission channels’ delay, the neighbor nodes’ location and residual energy, automatically select the best transmission route, and adapt the data transmission accuracy to ensure efficient and reliable data transmission in underwater wireless sensor networks.

(4) The simulation results show that the edge prediction algorithm can meet the accuracy requirements of underwater time series data monitoring, and the data transmission algorithm based on reinforcement learning can dynamically adapt to the changes in the transmission environment, which can greatly reduce the data transmission delay and energy consumption of underwater wireless sensor networks, and effectively improve the packet-sending success rate.

The remainder of this paper is organized as follows: Section 2 introduces the related research work; Section 3 introduces the framework of the edge prediction-based adaptive data transmission algorithm (EP-ADTA); Section 4 introduces the edge prediction algorithm based on ARMA (EP-ARMA); Section 5 introduces the data transmission algorithm based on reinforcement learning (RL-ADTA); Section 6 introduces the simulation experiment and performance analysis; Section 7 is the conclusion.

## 2. Related Works

The complexity and time variation of the underwater network environment have encouraged researchers to study efficient and reliable data transmission methods from different perspectives. The routing protocol is the key to realizing data transmission in underwater wireless sensor networks. According to the implementation method, the routing protocol is divided into the geographic routing protocol [33,34] and the opportunistic routing protocol [7,16]. The geographic routing protocol uses the relative or absolute location information of underwater nodes to design the packet-forwarding path, avoiding a large amount of broadcast information caused by the routing search [35]. The opportunistic routing protocol solves the problem of intermittent connection caused by mobility and the dynamic underwater acoustic link in sparse acoustic wireless networks, and uses ‘store-carry-forward’ opportunistic routing to maximize packet forwarding. According to the guaranteed type of quality of services (QoS), routing protocols are divided into those that focus on energy consumption balance, transmission delay, network life, or other indicators. Routing protocols can also be divided into traditional routing protocols and intelligent routing protocols according to whether artificial intelligence algorithms are used [3,36]. Traditional routing protocols do not need data learning and model optimization. In specific network scenarios, the optimization effect of specific indicators is good, but the network adaptability is poor, and there is only one optimization goal. The intelligent routing protocol can collect the network status data, form awareness of the network environment through data learning, support the dynamic adjustment of routing, and realize the balanced optimization of multiple QoS indicators, and has strong network adaptability [36,37]. Researchers have also adopted a cross-layer design method to further improve the performance of the routing protocol through the joint design of data link layer, transport layer, and application layer protocols [10,38]. Several typical data transmission algorithms and their characteristics are listed in Table 1. The communication efficiency mainly reflects the ability of the routing protocol to send effective information and control the transmission delay and energy consumption.

RLOR uses opportunistic routing to improve the reliability of routing. The basis of RLOR’s selection of relay nodes is the use of transmission direction and distance, channel quality, number of neighbors, and residual energy of relay nodes to ensure that the best route can be dynamically selected. RCAR takes more account of the congestion avoidance method of relay nodes. In RCAR, the transmission delay and energy consumption distribution in underwater data communication are optimized. MCR-UWSN divides the network into multiple clusters; dynamically selects cluster heads based on the optimized heuristic algorithm; and constructs inter-cluster routes to achieve efficient data transmission. KACO adopts clustering and heuristic algorithms for network layering and clustering. PB-ACR can distinguish the priority of different applications. In PB-ACR, high- priority applications choose the shortest route, and low-priority applications choose the route with more residual energy. ACOR supports the adaptive adjustment of data sampling frequency according to data correlation to reduce transmission energy consumption. Although both RLOR and RCAR can provide good QoS under extremely limited transmission conditions, it is not enough to simply consider overcoming the problems of the underwater environment and seek the best transmission route. Dynamic network traffic will seriously affect the performance of routing protocols. These protocols also need to be combined with applications to achieve efficient information transmission, rather than just data transmission. The use of layers, clusters, and inter-cluster routing based on the heuristic algorithm has improved the balance of energy consumption of MCR-UWSN and KACO algorithms; however, there is no detailed description of the performance change in algorithms under different network traffic conditions. Although PB-ACR provides differentiated routing for different priority services, it does not reduce the overall energy consumption of the network. Although ACOR can improve communication efficiency by sampling service data, the sampling technology is not suitable for applications having a large number of irregular and abrupt data transmission requirements, and ACOR does not address the problem of abnormal data.

SDA is used for event detection and source determination. When the sensor’s sensing data are transmitted upward along the routing tree, SDA comprehensively determines whether the underwater network generates abnormal data by comparing the sensing data of multiple adjacent sensor nodes with the normal sensing range. The neighboring sensor nodes only collect and forward their sensing data to the receiving node when these sensor nodes jointly detect possible events. This strategy avoids the unnecessary routing of perceptual data that is unfavorable to event detection, reduces energy consumption, and increases network capacity. However, this fusion can save energy for the network only when it is used to judge abnormal conditions. DBP is an event detection method based on linear prediction. The core idea of DBP is to use a simple model to capture the trend in data changes, and use flexible rules to calculate and deal with the existing interference. One of the limitations of DBP is its low prediction accuracy, which is suitable for short-term linear trend data. Although both SDA and DBP are based on realizing the communication of sensing data and can also be applied to underwater 3D networks, they focus on the detection of events, and do not address the construction of transmission routes and the methods for delivering data from sensor nodes to sink nodes in detail.

TBDP is a two-level bidirectional prediction model that is suitable for 3D underwater wireless sensor networks with an AUV, and realizes communication by relying on the data carried by the AUV. TBDP defines the AUV auxiliary network as the end–edge–cloud system, in which the underwater sensor system including the sensor node and the communication gateway node is defined as the end system, the AUV and the water surface sink node are defined as the edge system, and the shore-based data center is defined as the cloud environment [43,44]. Two-way prediction algorithms are set between the end system and the edge system, and between the edge system and the cloud environment, to compress the transmission data demand. Although using an AUV as a method of underwater data acquisition has certain advantages, the real-time performance of data acquisition is still insufficient compared with that of the fixed 3D underwater sensor network. In addition, the prediction algorithm proposed by TBDP has room for improvement in prediction accuracy, applicability, and mobility for different application requirements.

Therefore, on the basis of the above research, in order to improve the communication efficiency of the fixed 3D underwater wireless sensor network, it is necessary to carry out research on the adaptive data transmission algorithm that comprehensively considers business applications and the transmission environment. The algorithm realizes data compression or preprocessing based on edge prediction, and realizes the selection of relay nodes and the control of data transmission accuracy based on reinforcement learning.

Underwater environment monitoring includes the monitoring of the underwater temperature, salinity, pressure, PH value, dissolved oxygen, NH4-N, and other time series data, and various applications have great differences in terms of the quantity, accuracy, frequency, speed, and reliability of data transmission. However, in this paper, only the underwater temperature monitoring is taken as an example to illustrate the system framework and algorithm. Other types of monitoring applications can be improved in the system framework in combination with the application characteristics.

## 3. System Architecture

### 3.1. End–Edge–Cloud Architecture

The fixed 3D underwater wireless sensor network is composed of shore-based/ship data centers, buoy sink nodes, underwater floating sensor/communication nodes, and bottomed sink sensor nodes [3,45]. Both the bottom and floating sensor nodes have data acquisition capabilities. Based on the wireless communication mode, the data are sent to the underwater communication nodes, and then forwarded via each hop through the underwater communication nodes to send the monitoring data to the surface sink nodes, and then sent by the sink nodes to the data center via satellite. End–edge–cloud architectures include the cloud environment, edge network, and information end. In the Internet of Things, the information terminal is distributed in the monitoring environment and is responsible for data collection; the edge network provides data access channels for various information terminals and is responsible for data transmission; and the cloud environment gathers all kinds of data, is responsible for centralized data processing, and provides services for upper layer applications. Therefore, the sensor nodes responsible for data collection are defined as the information end; the underwater wireless communication network composed of the underwater sensor, communication, and sink nodes is defined as the edge network; and the broadband wireless communication network composed of shore- and ship-based data centers is uniformly defined as the cloud environment; this is shown in Figure 1.

In the underwater monitoring application, the data obtained by the information terminal are transmitted via each hop through the edge network and continuously sent to the cloud environment. The change trend, acquisition accuracy, and outliers of monitoring data are very important for monitoring applications, but the huge quantity of transmitted data will pose large problems for the edge network. As the delay increases, the energy consumption increases, and the packet-sending success rate decreases. Therefore, before the data enters the edge network, the network throughput should be reduced and the data transmission performance improved by preprocessing to realize the transmission data compression. Although data preprocessing must allocate the computational energy consumption for underwater network nodes, the computational energy consumption is often far less than the communication energy consumption. According to [46], the energy consumed by executing 3000 instructions is equivalent to transferring 1 bit of data a distance of 100 m. With the rapid improvement of computing chip technology and computing modes [47], the gap between computing energy consumption and communication energy consumption will continue to widen.

The data collected by underwater three-dimensional monitoring belongs to time series data, which have certain continuity in space and time, and it is possible to reduce the amount of data transmission through preprocessing. There are two methods to realize front-end monitoring data compression, i.e., using the adaptive sampling method or the time series data prediction algorithm. The frequency of adaptive sampling is closely related to the frequency of sampling data. If the sampling frequency is small, key data will be missed. If the sampling frequency is large, the effect of data compression will be lost. When the monitoring data changes dramatically in a short time, the adaptive sampling may not be able to capture the key data. Time series data prediction is based on historical data and data modeling to realize the learning and fitting of monitoring data, which can better reflect the change trend of the monitoring data. The parameters of the prediction model can be used to replace the actual monitoring data, and the data compression can be better realized. At the same time, selecting correction data of different threshold ranges can effectively adjust the accuracy of the monitoring application and avoid key data having drastic changes being missed. The surface sink node recovers the monitoring data according to the received prediction parameters and correction data based on the edge prediction algorithm.

### 3.2. System Composition

The nodes in the fixed 3D underwater wireless sensor network have certain computing and storage capabilities. Based on the underwater acoustic mode, they receive and send service data, and learn the status of the underwater transmission environment by monitoring the communication of the neighbor nodes. The monitoring data are collected from the sensor nodes, and forwarded by the communication nodes to the sink nodes. Therefore, each node constituting the edge network can be defined according to the agent, as shown in Figure 2.

The agent in the edge network includes the communication module, edge prediction module, and intelligent data transmission module, and has an application requirements base, environment status base, transmission policy base, and service data base. The edge prediction module is only required for underwater sensor nodes and sink nodes. The application requirements base stores the requirements for data collection type, frequency, and accuracy of monitoring applications. The environment status base stores the state information of neighbor nodes, transmission channels, and the current node. The transmission policy base stores super-parameters, thresholds, and the V values [48]. The service data base stores the monitoring data, newly received prediction parameters, correction data, etc., and temporarily stores the sent data. The communication module is responsible for data transmission and receiving the packets. The edge prediction module can not only automatically match the optimal prediction parameters, but also generate monitoring prediction data based on the prediction model. The intelligent data transmission module is responsible for selecting the appropriate transmission route and data transmission accuracy according to the status of the environment. When the status update of the neighbor node is detected, the communication module is responsible for updating the content of the environment status base. When new data are received, the communication module receives the data; obtains the prediction parameters, correction values, and data source node information; and writes the new data into the service data base. When sending data, the intelligent data transmission module generates the best data transmission strategy and sends the data according to the selected route and data accuracy. When the communication module finds that the neighbor node has sent data, the intelligent data transmission module deletes the temporary data in the service database.

### 3.3. Operation Flow

In order to achieve efficient and reliable data transmission in underwater wireless sensor networks, cooperation between agents located in the edge networks is required. According to the location and function of the agent in the edge network, the agents are divided into the sensor agent, communication agent, and sink agent, corresponding to the underwater sensor node, communication node, and sink node, respectively.

The sensor agent should complete the following actions: (1) carry out data acquisition and edge prediction, and generate prediction parameters and correction data; (2) assemble data packets, generate adaptive transmission policies, and send the packets to relay nodes according to the policies; and (3) receive the packets and update the status of transmission environment.

The communication agent should complete the following actions: (1) repackage the data, generate an adaptive transmission strategy, and send the packet to the relay node according to the strategy; (2) receive the packets, update the status of transmission environment, and obtain the data.

The sink agent should complete the following actions: (1) receive data packets and obtain transmission data; (2) generate the new monitoring data based on edge prediction, historical data, received prediction parameters, and correction data, and then send them to the cloud environment.

The functions of the sensor agent, communication agent, and sink agent are composed of four basic processes, namely: (1) the data acquisition and edge prediction process; (2) the data transmission process; (3) the data receiving process; (4) and the edge prediction and data recovery process. Of these, the sensor agent needs to implement processes (1), (2) and (3); the communication agent needs to implement processes (2) and (3); and the sink agent needs to implement processes (3) and (4).

Process (1) is shown in Figure 3a. Based on the ARMA [49] model, it can automatically match the optimal prediction parameters, generate prediction parameters and correction data according to historical data, and provide the threshold of the maximum allowable error for the monitoring application.

Process (2) is shown in Figure 3b. The reinforcement learning calculates the new Q value [48] according to the status of neighbor nodes, transmission channels, and their own nodes, in addition to the V value, threshold and super-parameters, and finds the best transmission strategy, that is, the next hop node and the number of correction data. Then, the packets are generated according to the packet format and broadcast. When the sender hears that the data packet has been forwarded by the neighbor node, the sent data are deleted; otherwise, the Q value is calculated and retransmitted.

Process (3) is shown in Figure 3c. By monitoring and receiving the new data in the transmission environment, the status of the neighbor nodes and the transmission channels are updated. If this node is a relay node, the new data are received.

Process (4) is shown in Figure 3d. It is only used in the sink node and is implemented based on the ARMA model. It is used to restore the generated monitoring data according to the received prediction parameters, historical data, and correction parameters, and to send the generated monitoring data to the cloud environment.

Next, we introduce the edge prediction algorithm based on EP-ARMA and the adaptive data transmission algorithm based on RL-ADTA.

## 4. Edge Prediction Algorithm Based on ARMA

### 4.1. ARMA Model

A time series refers to the results of observing a certain process at a given sampling rate in an equally spaced time period [49]. Environmental monitoring data such as temperature and salinity in the application of underwater three-dimensional monitoring are a form of time series data. The change in the time series has a tendency, periodicity, and seasonality. The change process of the time series can be defined as a set of variables: $y_{t}$ is dependent on the parameter t. The core of edge prediction is time series prediction, that is, to find out the change law with time of underwater environmental data, such as temperature and salinity, from historical data, and to predict future data.

Considering the computing performance of underwater nodes and the prediction accuracy of underwater monitoring applications, the ARMA model is used as the core model of edge prediction. ARMA is often used for stationary time series prediction [49]. If the monitoring data are a stationary series, the monitoring data can be expressed by the combination of pre-sequence data and prediction error, and the expression is as shown in [49]:(1)$$y_{t}=\beta_{0}+\beta_{1}y_{t-1}+\cdots +\beta_{p}y_{t-p}+\varepsilon_{t}+\alpha_{1}\varepsilon_{t-1}+\cdots +\alpha_{q}\varepsilon_{t-q},\;$$
where y is the predicted or observed values, and ε indicates the prediction errors; α and β are the coefficients; p and q are the orders, indicating the number of the observed values and prediction errors, respectively. Formula (1) represents time series $y_{t}$ following the ARMA(p,q) model.

When the monitoring data are non-stationary series, that is to say, the time series contains the characteristics of trend, seasonality and periodicity, the nonstationary sequence can be converted into a stationary sequence through difference processing. The second-order difference expression is as shown in [49]:(2)$$\nabla^{2}y_{t}=\nabla \left(y_{t}-{\;y}_{t-1}\right){=y}_{t}-{\;2y}_{t-1\;}{+y}_{t-2},\;$$
where ∇ is the difference operator, and $\nabla^{2}$ is the second-order difference operator. The difference reflects the change between discrete quantities. The first-order difference represents the difference between the current value and the previous value, and the second-order difference represents the difference between the current first-order difference and the previous first-order difference. Formula (2) represents the second-order difference of time series $y_{t}$.

$\nabla^{d}y_{t}$ is the d-order difference of $y_{t}$ After the difference transformation, $\nabla^{d}y_{t}$ becomes a stationary time series, which can be expressed as shown in [49]:(3)$$\lambda \left(B\right)\left(\nabla^{d}y_{t}\right)=\theta \left(B\right)\varepsilon_{t},\;$$
where B is the delay operator, which follows $y_{t-p}=B^{p}y_{t},\;\left(\forall p\geq 1\right)$ and $\nabla^{k}={\left(1-B\right)}^{k}$. $\lambda \left(B\right)=1-\lambda_{1}B-\lambda_{2}B^{2}-\cdots -\lambda_{p}B^{p}$ is the autoregressive polynomial, and $\theta \left(B\right)=1-\theta_{1}B-\theta_{2}B^{2}-\cdots -\theta_{q}B^{q}$ is the moving average polynomial. $\varepsilon_{t}$ is a white noise sequence of the zero-mean. Formula (3) is also called the ARIMA(p,d,q) model, which is used to realize edge prediction.

The autocorrelation function (ACF) is often used to determine the order d of the ARIMA model [50,51,52]. After d-order difference transformation, if the ACF value corresponding to the difference series is close to 0, the time series passes the stationarity test. Smaller values of d represent a better choice.

When selecting the p and q parameters, the Akaike information criterion (AIC) and Bayesian information criterion (BIC) can achieve a better selection effect than using the mean square error (MSE) [53]. AIC is based on the concept of entropy and provides a standard to weigh the complexity of the estimation model and the goodness of the fitting data. AIC is defined in [53] as follows:(4)AIC=−2log(L)+2(p+q+k+1), 
where L is the likelihood function of the monitoring data; p and q represent the parameters of the ARMA model, respectively; and k is taken as 1 or 0, indicating whether the error term is considered. When there is a large difference between the two models, the difference is mainly reflected in the likelihood function term. When the likelihood function difference is not significant, the second term of Formula (4), that is, the model complexity, plays a role, so the model with fewer parameters can be selected.

BIC can also realize model selection. BIC is defined in [53] as follows:(5)BIC=AIC+[log(T)− 2](p+q+k+1), 
where T is the number of the monitoring samples. The ARMA (p, q) model with smaller AIC and BIC values not only has a better fitting degree, but also has fewer parameters. The model with fewer parameters can effectively reduce the over-fitting degree of the model.

In this paper, an effective method of automatically traversing the model space is used to select the parameters of the ARIMA(p,d,q) model. The method can select the best d-order, and the p and q parameters according to AIC or BIC at the same time. The method first tries to select several possible parameter models, and then fine tunes the parameters according to the prediction results to quickly match the model parameters of ARIMA(p,d,q) [53].

### 4.2. EP-ARMA Implementation

The sensor nodes will collect monitoring data sets as $\left\{{\mathrm{collect}}_{i}\right\}$ in the monitoring period T. Additionally, then, based on the ARIMA(p,d,q) model and historical data, the forecast data sets $\left\{{\mathrm{forecast}}_{i}\right\}$ can also be generated, where i is the time point of data collection. When the underwater monitoring data acquisition accuracy is {accuracy}, the following requirements should be met to ensure the result of edge prediction is suitable for underwater monitoring:(6)$$\forall \;i\in \left\{1,\cdots ,T\right\}\left|{\mathrm{forecast}}_{i}-{\mathrm{collect}}_{i}\right|\leq \mathrm{accuracy},\;$$

However, due to the randomness and abruptness of time series changes, it is not possible to ensure that all prediction data can fulfil Formula (6). Therefore, a certain amount of collected data should be used as the correction data. When there is a large deviation between the predicted data and the collected data at a certain time point, the collected data at that time point are used to replace the prediction data, and the corrected prediction data set $\left\{{\mathrm{forecast}}_{i}^{\prime }\right\}$ and the correction data set $\left\{n,i,{\;\mathrm{correct}}_{j}\right\}$ are generated, where n is the number of correction data, i is the correction time point, and *j* is the identification of correction data. By transmitting the prediction parameters and the correction data set, the corrected prediction data set $\left\{{\mathrm{forecast}}_{i}^{\prime }\right\}$ that meets the accuracy requirements of Formula (6) can be recovered at the remote end.

The underwater monitoring data acquisition accuracy is dynamically adjusted within the range of $\left\{{\mathrm{accuracy}}_{\mathrm{high}},\cdots ,{\;\mathrm{accuracy}}_{\mathrm{low}}\right\}$. When the number of correction data sets n is equal to $n_{min}$, the corrected forecast data set $\left\{{\mathrm{forecast}}_{i}^{\prime }\right\}\;$ can meet the ${\mathrm{accuracy}}_{\mathrm{low}}$. When the number of correction data sets is equal to $n_{\max}$, the corrected forecast data set $\left\{{\mathrm{forecast}}_{i}^{\prime }\right\}$ can meet the ${\mathrm{accuracy}}_{\mathrm{high}}$. Therefore, n will change in $\left\{n_{\min},\cdots ,{\;n}_{\max}\right\}$. The prediction parameters (p, d, q) and correction time points i are also transmitted when the correction data are transmitted. If the number of correction data sets is taken as $n_{\min}$, the amount of data to be transmitted is still greater than the amount of data to be collected directly, and the algorithm will not be applicable. Assume that p, d, q and i are expressed as integer data, and the collected and correction data are expressed as float data. Then, 1 byte represents the integer data, 4 bytes represent the float data, and when $n_{\min}$ cannot meet:(7)$$n_{\min}<\frac{4\times T-3}{5},\;$$

EP-ARMA will not be suitable for underwater monitoring applications with such accuracy. It is more efficient to directly transmit the collected data or adjust the prediction model. When $n_{\min}$ fulfils Formula (7), the number of correction data sets can be adjusted to reduce the amount of data actually transmitted in the underwater network and improve the efficiency of underwater monitoring data transmission. $n_{\min}$ is set as the threshold value. Each node needs to ensure that the number of transmission correction nodes is not less than the threshold value.

The value of ${\mathrm{accuracy}}_{\mathrm{low}}$ is generally given by the application program, which is an insurmountable threshold for data transmission. The ${\mathrm{accuracy}}_{\mathrm{low}}$ is constant in the whole communication process. The value of ${\mathrm{accuracy}}_{\mathrm{high}}$ is initially given by the application, but is limited to Formula (7) and the prediction result, and will change dynamically according to the amount of correction data allowed to be transmitted. The value of ${\mathrm{accuracy}}_{\mathrm{high}}$ remains constant or decreases in the whole communication process. $\left\{n_{\min},\cdots ,{\;n}_{\max}\right\}$ will change dynamically with $\left\{{\mathrm{accuracy}}_{\mathrm{high}},\cdots ,{\;\mathrm{accuracy}}_{\mathrm{low}}\right\}$, where $n_{\min}$ represents the lowest threshold allowed by the service accuracy, and $n_{\max}$ represents the highest threshold that the edge prediction algorithm can provide. $\left\{n_{\min},\cdots ,{\;n}_{\max}\right\}$ and $\left\{{\mathrm{accuracy}}_{\mathrm{high}},\cdots ,{\;\mathrm{accuracy}}_{\mathrm{low}}\right\}$ play a consistent role in EP-ARMA.

EP-ARMA, as shown in Algorithm 1, are generally deployed in sensor agents and sink agents. They must find the {p,d,q} parameters with the lowest AIC or BIC value according to the automatic traversal algorithm. Additionally, it generates the prediction data set $\left\{{\mathrm{forecast}}_{i}\right\}$. Then, it generates the correction data set $\left\{n,i,{\;\mathrm{correct}}_{j}\right\}$ and the threshold value. The edge prediction on the sink agent recovers the corrected prediction data set $\left\{{\mathrm{forecast}}_{i}^{\prime }\right\}$, according to {p,d,q}, $\left\{n,i,{\;\mathrm{correct}}_{j}\right\}$, the threshold value, and the historical data.
**Algorithm 1** Edge Prediction Based on ARMA (EP-ARMA)Initialize p, d, q range, $\left\{{\mathrm{accuracy}}_{\mathrm{high}},\ldots ,{\;\mathrm{accuracy}}_{\mathrm{low}}\right\}$, forecast step, T Complete the monitoring task in period T, and obtain the data of $\left\{{\mathrm{collect}}_{i}\right\}$
While (true)   If (node.type is sensor)    According to the forecast step, obtain the ${\mathrm{data}}_{\mathrm{history}}$ from ‘service data’ database    Obtain the ARIMA($p_{\mathrm{best}}$, $d_{\mathrm{best}}$, $q_{\mathrm{best}}$) by AIC or BIC in p, d, q range    Obtain the $\left\{{\mathrm{forecast}}_{i}\right\}$ by ARIMA($p_{\mathrm{best}}$, $d_{\mathrm{best}}$, $q_{\mathrm{best}}$), ${\mathrm{data}}_{\mathrm{history}}\;$ and T    Obtain the $\left\{n_{\min},\cdots ,{\;n}_{\max}\right\}$, $\left\{n,i,{\;\mathrm{correct}}_{j}\right\}$, and threshold by $\left\{{\mathrm{collect}}_{i}\right\}$, $\left\{{\mathrm{forcast}}_{i}\right\}$, $\left\{{\mathrm{accuracy}}_{\mathrm{high}},\cdots \;,{\mathrm{accuracy}}_{\mathrm{low}}\right\}$ and Formula (7)    Save the ($p_{\mathrm{best}}$, $d_{\mathrm{best}}$, $q_{\mathrm{best}}$) and $\left\{n,i,{\;\mathrm{correct}}_{j}\right\}$ in ‘service data’ database    Save the ‘threshold’ in ‘transport policy’ database    Wait for the next period T   End if  If (node.type is sink && the new data has been received)    According to the forecast step, get the ${\mathrm{data}}_{\mathrm{history}}$ from ‘service data’ database    Obtain the (p, d, q), $\left\{n,i,{\;\mathrm{correct}}_{j}\right\}$ and threshold from the received data    Obtain the $\left\{{\mathrm{forecast}}_{i}^{\prime }\right\}$ by ARIMA (p, d, q) and $\left\{n,i,{\;\mathrm{correct}}_{j}\right\}$
   Save the $\left\{{\mathrm{forecast}}_{i}^{\prime }\right\}$ in ‘service data’ database   End if end while

## 5. Adaptive Data Transmission Algorithm Based on Reinforcement Learning

### 5.1. Reinforcement Learning Model

Reinforcement learning is a kind of learning that aims to obtain the best mapping strategy from environmental state to action. The best strategy can maximize the reward, which is the evaluation of the movement quality. Due to the limited information provided by the external environment, reinforcement learning can improve the ability to adapt to the dynamic environment by continuously learning and acquiring feedback knowledge in the process of action evaluation. Reinforcement learning algorithms are often described by Markov decision processes (MDPs). The process of reinforcement learning adopts the five tuples of (S, A,P,R,γ), where S stands for environment status, A stands for action, P stands for transition probability, R stands for reward, and γ represents the discount rate. Agents, according to the environment status S, cumulative reward R, and discount rate γ, determine the transition probability P, and generate action A, then switch to the new state  S ′ [48]. In underwater wireless sensor networks, the process of data transmission can also be regarded as Markov decision processes (MDPs). Reinforcement learning is used to select the best relay node and meet the required data transmission accuracy, as shown in Figure 4.

Assuming that the nodes of the underwater wireless sensor network can be expressed as follows:(8)$$N=\left\{n_{1},n_{2},n_{3},\cdots ,n_{m}\right\},\;$$
where *n* represents the node and *m* is the number of nodes, each node in the network can obtain its own location information. Therefore, a virtual pipeline can be built on the vector between the data source node and the target sink node, as shown in Figure 5. The virtual pipe radius of the node i is set to:(9)$$R_{\mathrm{pipe}}^{i}=-\frac{R-R_{\mathrm{ini}}^{i}}{E_{\mathrm{ini}}}\times \bar{E^{i}}+R\;,\;$$
where $E_{\mathrm{ini}}$ is the initial energy of the node, $\bar{E^{i}}$ is the average residual energy of the neighbor nodes, R is the signal transmission distance of the node, and $R_{\mathrm{ini}}^{i}$ is the initial radius of the pipeline. When the neighbor nodes have more residual energy, they can maintain a small pipeline radius, allowing the data to acquire the shortest path. When the neighbor nodes have less residual energy, they can enlarge the pipeline radius of transmission, allowing more neighbor nodes to participate in packet forwarding.

The candidate relay node set $N_{relay}\left(i\right)$ of the current node $n_{i}$, can be expressed as follows:(10)$$N_{\mathrm{relay}}\left(i\right)=\left\{n_{j}\in N_{i}\left(t\right)\subseteq \left\{N|\mathrm{dep}\left(n_{j}\right)-\mathrm{dep}\left(n_{i}\right)\leq 0\right\}\cap^{\;}\mathrm{neighbors}\left(n_{i}\right)\cap^{\;}\mathrm{pipe}\left(n_{i}\right)\right\}\;,\;$$

At time *t*, the node set of candidate relay nodes for $n_{i}$ is $N_{i}\left(t\right)$. The node set of neighbors overridden by the one-hop transmission of the node $n_{i}^{\prime }s\;$ signal is $\mathrm{neighbors}\left(n_{i}\right)$. The node set in the $n_{i}^{\prime }s$ virtual pipe is $\mathrm{pipe}\left(n_{i}\right)$. The node set with a shallower depth than the current node $n_{i}$ is $\left\{N|\mathrm{dep}\left(n_{j}\right)-\mathrm{dep}\left(n_{i}\right)\leq 0\right\}$. The relay node selection needs to be within the range defined by Formula (10).

**Definition** **1.***At time t, if the packet to be sent is at node* $n_{i}$*, then the current environment state S can be defined as follows:*(11)$$S=\underset{\mathrm{acc}}{\cup }\left(\left\{n_{i}\right\}\cup^{\;}N_{\mathrm{relay}}\left(i,\mathrm{acc}\right)\right),\;$$*where ‘acc’ denotes that the monitoring data are located in the variable accuracy range of the node, which is determined by*$\left\{{\mathrm{accuracy}}_{\mathrm{high}},\cdots ,{\;\mathrm{accuracy}}_{\mathrm{low}}\right\}$*or*$\left\{n_{\min},\cdots ,{\;n}_{\max}\right\}$. *The environment includes not only all possible relay nodes, but also all possible data accuracy on different nodes.*

**Definition** **2.**
*At time t, action A can be defined as follows:*

(12)
$$A=\left\{a_{j,\mathrm{acc}}|(n_{j}\in S,\;\mathrm{acc}\in \left\{n_{\min},\cdots ,{\;n}_{\max}\right\})\right\}\;,\;$$


*If the relay node completes data forwarding, it will be rewarded. If the relay node is closer to the water surface and has more residual energy, and the delay of transmission channel is smaller, it will be more conducive to the efficient transmission of data to the sink node, and it will be given additional rewards. If it adopts higher transmission accuracy and transmits more correction data, it will also receive additional rewards.*


**Definition** **3.***At time t, if the packet is at node* $n_{i}$*, and*$n_{i}$*selects*$n_{j}$*as a relay node, and sends the packet with the accuracy ‘acc’, the reward function is as follows:*(13)$$R_{n_{i}n_{j,\mathrm{acc}}}^{a_{j,\mathrm{acc}}}=-R_{0}-\left[\varphi_{e}\times \mathrm{co}\left(e\right)+\varphi_{t}\times \mathrm{co}\left(t\right)+\varphi_{l}\times \mathrm{co}\left(l\right)\right]\;,\;$$
*Part I: the fixed cost*

$R_{0}$

*will be accumulated with each data forwarding. The existence of*

$R_{0}$

*can help the agent select a route with fewer hops. However, the shortest route is not always the best route. It also needs to comprehensively consider the factors such as the nodes, channels, and data transmission accuracy.*

*Part II: *

co(e)

*is the cost for node energy. The existence of*

co(e)

*can help the agent select the node having larger residual energy as the relay node, which is more conducive to the continuation of the network lifecycle.*

co(e)

*can be expressed as follows:*

(14)
$$\mathrm{co}\left(e\right)=1-\frac{E_{\mathrm{res}}^{j}}{\sum_{k\in N_{\mathrm{relay}}\left(i\right)}E_{\mathrm{res}}^{k}}\;,\;$$

*where*

$E_{\mathrm{res}}^{j}$

*represents the residual energy of the next hop node,*

$\underset{k\in N_{\mathrm{relay}}\left(i\right)}{\sum }E_{\mathrm{res}}^{k}$

*represents the total remaining energy of the candidate node set, and*

$\varphi_{e}$

*is the sensitivity coefficient of residual energy cost of neighbor nodes.*
*Part III:*co(t)*is the cost for the delay of the transmission channel. The existence of*co(t)*can help the agent select the node having smaller transmission delay as the relay node. This means that the channel between the current node and the relay node is stable and reliable, has a low bit error rate and has less congestion.*co(t)*can be expressed as follows:*(15)$$\mathrm{co}\left(t\right)=1-\beta_{1}\frac{1}{\bar{t_{\mathrm{total}}^{i\to j}}+1}-\beta_{2}\frac{1}{\bar{t_{\mathrm{receive}}^{j}}+1}\;,\;$$*where*$\bar{t_{\mathrm{total}}^{i\to j}}$*is the average of total delay from node*$n_{i}$*to node*$n_{j}$*. The total delay refers to the time from*$n_{i}$*sending the data until*$n_{i}$*hears*$n_{j}$*forwarding the same data, which includes the sender delay in*$n_{i}$*, the propagation delay from*$n_{i}$*to*$n_{j}$*, the processing and waiting delay in*$n_{j}$*, and the propagation delay from*$n_{j}$*to*$n_{i}$. $t_{\mathrm{timeout}}$*is the maximum value of the total delay node*$n_{i}$*to node*$n_{j}$*, marking data transmission failure. It is generally defined according to the transmission conditions and business requirements*. $\bar{t_{\mathrm{receive}}^{j}}$*is the average transmission delay from neighbor node*$n_{j}$*to the current node*$n_{i}$*, which can be calculated by the time stamp of the packets from*$n_{j}$. $\varphi_{t}$*is the sensitivity coefficient of the transmission channel delay cost.*$\beta_{1}{\;\mathrm{and}\;\beta }_{2}$*are used to adjust the weight of the two delay parameters, respectively.*$\bar{t_{\mathrm{total}}^{i\to j}}\;$*is the actual delay of packet transmission.*$\bar{t_{\mathrm{receive}}^{j}}$*can help nodes to optimize transmission policies without forwarding the packets.*
*Part IV:*

co(l)

*is the cost related to data transmission accuracy. The data transmission accuracy is determined by the number of correction data transmitted.*

co(l)

*can be expressed as follows:*

(16)
$$\mathrm{co}\left(l\right)=\frac{1}{l_{\mathrm{correct}}+1}\;,\;$$

$l_{\mathrm{correct}}$*indicates the number of correction data sent. The value range of*$l_{\mathrm{correct}}$*is*$\left[\mathrm{threshold},\cdots \;,l_{\mathrm{correct}}^{\max}\right]$. $l_{\mathrm{correct}}^{\max}$*is the number of correction data received or generated by the node*$n_{i}$. $\varphi_{l}$*is the sensitivity coefficient of the correction accuracy cost. Selecting higher*$\varphi_{l}$*can reduce the accuracy of the monitoring data. However, the greater length of the transmitted data can reduce the packet-sending success rate, and increase the transmission delay. Although selecting lower*$\varphi_{l}$*will reduce the accuracy of data transmission, it can improve the reliability of data transmission. Therefore, selecting the appropriate*$\varphi_{l}$*will realize the win–win of transmission efficiency and transmission accuracy.*

**Definition** **4.***At time t, if the current packet is in node* $n_{i}$*, the transition probability of node*$n_{i}$*to node*$n_{j}$*the accuracy ‘acc’ is defined as follows:*(17)$$P_{n_{i}n_{j,\mathrm{acc}}}^{a_{j,\mathrm{acc}}}=\frac{R_{n_{i}n_{j,\mathrm{acc}}}^{a_{j,\mathrm{acc}}}}{\sum_{n_{k,{\mathrm{acc}}^{\prime }}\in S}R_{n_{i}n_{k,{\mathrm{acc}}^{\prime }}}^{a_{k}}}\;,\;$$*In order to reflect the impact of the future status on the current status, the overall reward at time t is defined as shown in* [5]*:*(18)$$R_{t}=r_{t}+{\gamma r}_{t+1}+\gamma^{2}r_{t+2}+\cdots =\overset{\infty }{\underset{j=0}{\sum }}\gamma^{j}r_{t+j}\;,\;$$
*when* γ=0*, the system will only consider the reward r of the current action, not the future status. When*
γ→1*, the system will consider both the current and future action rewards r.**According to Q-learning, the state action function under policy*π*is defined as shown in* [5]*:*(19)$$Q^{\pi }\left(s,a\right)=E_{\pi }\left\{R_{t}|s_{t}=s,a_{t}=a\right\}\;,\;$$*Assuming that* a ′*and* s ′*are the next action and state, the optimal solution*$Q^{*}\left(s,a\right)$*in the state of*$Q^{\pi }\left(s,a\right)$*can be expressed as an iterative equation, as shown in* [5]*:*(20)$$Q^{*}\left(s,a\right)=r_{t}+\gamma \sum_{\;s\;\prime \in S}P_{s\;s\;\prime }^{a}\left\{\underset{\;a\;\prime }{\max}Q^{*}\left(\;s\;\prime ,\;a\;\prime \right)\right\}\;,\;$$
*According to the state action function, the node with sufficient residual energy and smaller transmission channel delay in the virtual pipeline will be selected as the relay node. The packets will be packaged according to the appropriate transmission accuracy meeting the threshold requirements and sent by the current node.*
*At time t, the value function V will select the state action function that can obtain the maximum benefit, which is defined as shown in* [5]*:*(21)$$V_{t}^{*}\left(s\right)=\underset{a}{\max}Q^{*}\left(s,a\right)\;,\;$$
*At the initial stage of reinforcement learning, the initial V value of each agent is 0. With the accumulation of learning, the V value will be continuously updated according to Formulas (20) and (21) and gradually converge. A good data forwarding strategy will finally emerge.*


### 5.2. RL-ADTA Implementation

RL-ADTA is implemented, as shown in Algorithm 2. The agents in underwater wireless sensor networks can obtain the status information of neighbor nodes and transmission channels by monitoring the packet information transmitted in the network. When a new packet is generated or received, the Q value is calculated according to the environment state of the current node where the packet is located, and the best policy is selected to send the packet. Once the packet is sent, the agent will record whether the packet is successfully sent, and update the agent state.
**Algorithm 2** Adaptive Data Transmission Algorithm Based on Reinforcement Learning (RL-ADTA)Initialize the positions of the nodes While (true)    If (the new data to send)     While (${\mathrm{times}}_{\mathrm{send}}<{\mathrm{times}}_{\max}$)      Create the virtual routing pipe using Formula (9)      Obtain the status of neighbors, channels, and the current node from the environment status base      Obtain the V-value, threshold, and super-parameters from the transport policy base      Update the S and A using Formulas (10)–(12)      Calculate Q function using Formulas (13)–(20)      Update the V-value with the max *Q* value using Formula (21)      Determine the number of correction data sent $n_{\mathrm{new}}$ by the max Q value      Determine the relay node using the max Q value      Form the packet by (p, d, q), $\left\{n_{\mathrm{new}},i,{\;\mathrm{correct}}_{j}\right\}$ and V-value      Forward the packet to the relay node      If (It is detected that the packet has been forwarded)       Break      Else       ${\mathrm{times}}_{\mathrm{send}}$++      End If     End While    End if End While

### 5.3. RL-ADTA Packet Design

The data transmission algorithm between the nodes is finally realized by transferring the packets containing the control and application information. RL-ADTA does not generate special requirements for the lower layer protocol, nor does it design data packets separately to transfer status or control information between nodes. The packet defined by RL-ADTA contains both the service information and the node status information. The underwater network nodes update the environment status by monitoring the packets sent by the neighbor nodes. When the relay node is the current node, the optimal data forwarding strategy is obtained according to RL-ADTA, and the packet is repackaged and forwarded to the next node. The packet structure of RL-ADTA is shown in Figure 6. The data package contains two parts: the fields in the ‘data head’ are related to underwater communication, while the fields in the ‘data content’ are related to monitoring applications.

The underwater monitoring application forms the monitoring data into data packets during every period T. The ‘data ID’ is the ID of the current period data, and the ‘destination address’ is the ID of the destination sink node. The ‘node ID’, ‘V value’, ‘residual energy’ of the current node, the ‘timestamp’ of sending data, and the ‘next forward’ representing the relay node ID are all included in the data header. The fields ‘task type ‘ and ‘threshold’ in the data content are determined by the type of monitoring data collected. Additionally, (p, d, q),’correction point’, $\left\{{\mathrm{correct}}_{1},\cdots \;,{\mathrm{correct}}_{\mathrm{num}}\right\}$ and ‘num’ are the edge prediction parameters, correction time-point, correction data, and their quantities, respectively.

## 6. Result and Discuss

### 6.1. Experimental Environment and Data

The experiment was based on the application of underwater temperature monitoring. The average depth of the ocean is within 3000 m. At the same water depth, the seawater temperature in the range of 5000 m × 5000 m is generally considered to be similar. Therefore, in the experiment, the range of underwater network was set to 5000 m × 5000 m × 2500 m. The transmission medium adopts an acoustic channel. The transmission speed of sound in water is affected by depth, temperature, and salinity. In the experiment, the sound velocity in seawater was set to 1500 m/s according to the common setting. The distance of sound propagation is related to its frequency. The communication distance of medium distance underwater acoustic communication is generally 1–10 km, and the bandwidth is in the order of 10 kHz. In the experiment, the frequency of sound was set to 10 kHz, and the communication distance and sensor distance were set to 1000 m, while the initial width of the virtual pipe was set to half of the communication distance. The adaptability of the algorithm should be verified by changing the density of different nodes. Therefore, in the experiment, 100, 200, or 300 nodes were selected to participate in the experiment. The initial energy of each node is 1000 J, the transmission power of the node is 10 W, the receiving power of the node is 3 W, the calculation energy power of the node is 48 mW, and the idle power of the node is 30 mW. The experimental data are derived from the temperature data observed 400 m underwater at the KEO station in the database of the National Oceanic and Atmospheric Administration (NOAA). In the data transmission experiment, the temperature data collected every 6 h are formed into data packets for transmission, the size of the original data packets is 50 bytes, and the transmission rate of the application is 1 kbps. In order to shorten the simulation process, the sending rate of the packet is 0.1 packet/s, and the simulation time is 2000 s. The temperature accuracy of 0.01, 0.1, and 0.2 °C was selected as the lowest accuracy allowed for application. All parameter settings of the experiment are displayed in Table 2.

The simulation environment was built based on Python 3.9, and the EP-ARMA and RL-ADTA algorithms were also implemented based on Python 3.9. The simulation environment comprised a computer having an i7-9700k CPU with a dominant frequency of 3.6 GHz, 64 GB memory, and GT1080 GPU.

The data are the underwater temperature data at 400 m below the KEO station from 12:00:00 on 16 June 2004 to 11:00:00 on 23 November 2004, i.e., a total of 3840 periods, as shown in Figure 7.

In order to verify the stability of the experimental data, the data distribution is formed after the first-order transformation, as shown in Figure 8.

The ADF of the original experimental data is 0.49, and the *p*-value is −1.59, which is greater than the probability (−2.56) of 10%, indicating that the data are unstable. After the first-order difference of the experimental data, the corresponding ADF is 1.25 × 10^−14^, close to 0, and the *p*-value is −8.89, which is far lower than the probability of 1% (−3.43), indicating that the data of the first-order difference tend to be stable.

### 6.2. Data Prediction Performance Analysis

#### 6.2.1. Prediction Accuracy Analysis

The data of KEO station are collected every hour. The 40-day (960 h) temperature data of four periods are used to predict the data of the next day (24 h) through the ARMA and EP-ARMA algorithms, respectively. The ARMA algorithm adopts the BIC value to automatically select the ARMA parameters. The lowest temperature accuracy of the EP-ARMA algorithm is 0.1 °C. The prediction performance analysis data are shown in Figure 9.

The left parts, (a), (c), (e) and (g), of Figure 9 show that the ARMA model based on the automatic optimization of parameters through BIC can indeed accurately use the historical data of the first 960 periods to predict the change trend of the temperature data in the next 24 h, but there are still some errors at each time point, especially at some key points with drastic changes. Although the EP-ARMA (0.1) model only corrects the data whose error between the predicted value and the monitored value exceeds 0.1 °C based on the prediction results of the ARMA model, it is shown by (b), (d), (f) and (h) on the right side of Figure 9 that the corrected data can accurately reflect the temperature data of the next 24 h in the four periods, and the key data with sharp changes in the period can also be extracted. Even in the face of temperature data in different periods, EP-ARMA can achieve more accurate prediction, especially to ensure that abnormal temperature changes are not omitted. Therefore, EP-ARMA can meet the requirements of data transmission accuracy for underwater monitoring applications, and this prediction ability is robust.

#### 6.2.2. Prediction Parameter Analysis

In time series analysis and prediction, the mean absolute error (MAE), mean square error (MSE), and root mean square error (RMSE) can calculate the deviation between the predicted value and the original value. The larger the deviation, the larger the MAE, MSE, and RMSE; the smaller the deviation, the smaller the MAE, MSE, and RMSE. Therefore, the MAE, MSE, and RMSE are often used to evaluate the performance of prediction algorithms [54]. Among these, the MAE is defined as shown in [54]:(22)$$\mathrm{MAE}=\frac{1}{N}\sum_{x=1}^{N}\left|p_{x}-{\hat{p}}_{x}\right|\;,\;$$

The MSE is defined as shown in [54]:(23)$$\mathrm{MSE}=\frac{1}{N}\sum_{x=1}^{N}{\left(p_{x}-{\hat{p}}_{x}\right)}^{2},\;$$

The RMSE is defined as shown in [54]:(24)$$\mathrm{RMSE}=\sqrt{\frac{1}{N}\sum_{x=1}^{N}{\left(p_{x}-{\hat{p}}_{x}\right)}^{2}},\;$$
where, the monitoring data are denoted by $p_{x}$, the forecast data are denoted by ${\hat{p}}_{x}$, the length of the data series is denoted by *N*, and *x* denotes the label of the data series.

Using the 40-day (960 h) temperature data in the four periods, ARMA models with different (p, d, q) parameters were selected to predict the data of the next day (24 h). The MAE, MSE, and RMSE curves calculated using the predicted data and the original monitoring data are shown in Figure 10.

Figure 10 shows that the curves of MAE, MSE and RMSE vary sharply with different (p, d, q) configurations. The curves in different periods show that the selection of different (p, d, q) values has an important impact on the prediction performance of EP-ARMA. The (p, d, q) configuration with the highest prediction accuracy is not completely consistent in different periods. Therefore, high-precision EP-ARMA prediction cannot be achieved with a fixed (p, d, q) configuration. It is necessary to find a new (p, d, q) configuration through the automatic parameter-matching method after a period of time, to ensure the stability of the predictive performance.

Figure 10 also shows that, due to the stability of the underwater temperature data, it is not necessary to change the (p, d, q) every period. The parameters in multiple consecutive periods do not change to a large extent. Therefore, it is only necessary to dynamically adjust the parameter of (p, d, q) when the prediction performance is greatly reduced. In the case of poor transmission conditions or stable changes in monitoring data, the data updating frequency can be reduced or even stopped. The subsequent data can be generated using EP-ARMA and historical data based on the (p, d, q) in the previous period, which can greatly improve the data transmission efficiency of the EP-ADTA.

#### 6.2.3. Forecast Model Analysis

Exponential smoothing (ES) is a classical time series prediction method [55]. The prediction generated by the exponential smoothing algorithm is the weighted average of the past observations. As the number of observations grows, the weighted values decay exponentially. This is suitable for short- and medium-term forecasting and has good accuracy. The prediction accuracy of the ES model depends on the smoothing factor α, which is set between 0.1 and 0.9. The smoothing factor α can be acquired using the trial-and-error method. In this experiment, the value of α is 0.22.

GRU is a variant of the RNN network and a type of long-term memory network that can effectively capture the nonlinear relationship between sequence data [56,57]. GRU adds a gating unit to the standard RNN model. The gating unit can control the flow status of information in different time steps in the network. GRU is a simplified version of the LSTM network, but the number of parameters used is still huge. The GRU structure used in the experiment adopts the structure of two layers of GRU and one layer of full connection, having 74,661 parameters.

The ES, ARMA, GRU, and EP-ARMA (0.1) models were used to predict the underwater temperature data based on three different training and prediction periods. The RMSE representing the performance of the four methods is shown in Figure 11.

Figure 11 shows that ES, ARMA, GRU, and EP-ARMA have similar prediction effects when forecasting with ‘168-4′ and ‘960-24′. ARMA, GRU and EP-ARMA have better prediction effects when forecasting with ‘1920-48′. Compared with GRU, ARMA and EP-ARMA have fewer parameters and smaller computational complexity, which is very important for resource-constrained underwater communication nodes. As the EP-ARMA model corrects the data, it gains a large return in accuracy; therefore, its RMSE is the lowest under the ‘168-4′, ‘960-24′, and ‘1920-48′ combinations. When selecting the prediction model, we should comprehensively consider the prediction performance and computing performance. Among the above four models, the EP-ARMA model is the best.

### 6.3. Transmission Efficiency Analysis

#### 6.3.1. Transmission Accuracy Analysis

According to the research of this paper, in the field of underwater wireless sensor networks, there is no algorithm that can dynamically adjust the accuracy of data transmission when selecting routes and sending data. Therefore, the existing state-of-the-art algorithms were not compared with the RL-ADTA in the analysis of the transmission accuracy. The deviation degree between the data collected and sent by the sensor node and the data received and recovered by the sink node is used to illustrate the performance of the RL-ADTA in controlling the accuracy of the transmitted data.

RL-ADTA (0.1, 0.2, 0.01) was used to carry out underwater temperature monitoring and data transmission experiments based on 20 different periods. The ${\mathrm{accuracy}}_{low}$ of RL-ADTA (0.1, 0.2, 0.01) determines the actual minimum amount of data to be transmitted, that is $n_{min}$ (0.1, 0.2, 0.01). The $n_{max}$ of RL-ADTA (0.1, 0.2, 0.01) are determined using Formula (7). When the monitoring data transmission interval is 6, the $n_{max}$ (0.1, 0.2, 0.01) are all set to be 4. The number of transmission correction data initially set by RL-ADTA (0.1, 0.2, 0.01) is 4, which represents the ${\mathrm{accuracy}}_{high}$ of RL-ADTA (0.1, 0.2, 0.01). During transmission, the accuracy of RL-ADTA (0.1, 0.2, 0.01) remains unchanged or gradually decreases according to the transmission environment, but the deviation is not lower than the ${\mathrm{accuracy}}_{low}$ (0.1, 0.2, 0.01).

After several rounds of iteration and training, the RL-ADTA (0.1, 0.2, 0.01) model converges. The data of 20 periods are sent into the underwater network transmission model based on RL-ADTA (0.1, 0.2, 0.01). The ratio between the data actually received by the sink node and the data that need to be sent by the sensor node, and the RMSE between the recovered data in sink node and the collected data in sensor node, are calculated and shown in Figure 12.

Figure 12 shows that, as the ${\mathrm{accuracy}}_{low}$ decreases, the RMSE of each period gradually decreases, but the number of data transmitted gradually increases. When the ${\mathrm{accuracy}}_{low}$ is 0.2, the mean value of the RMSE in 20 periods is 0.0903, the mean value of the received/sent ratio is 0.2583; and there is no need to send correction data in periods 7, 15, 16, 19, and 20. When the ${\mathrm{accuracy}}_{low}$ is 0.1, the mean value of the RMSE in 20 periods is 0.0473, the mean value of the received/sent ratio is 0.5292, and the number of sent correction data fluctuates greatly. When the selected threshold is 0.01, the mean value of the RMSE in 20 periods is 0.0286, the mean value of the received/sent ratio is 0.9021, and the number of the correction data actually received by the sink node is close to the number of the collected data that needs to be sent by sensor node. According to the data analysis—although the initial accuracy of RL-ADTA (0.1, 0.2, 0.01) is set to the same ${\mathrm{accuracy}}_{high}$—due to the difference in ${\mathrm{accuracy}}_{low}$ (0.1, 0.2, 0.01), and the fact that the parameter setting pays more attention to the packet-sending success rate and transmission efficiency, through multiple episodes of training, the transmission accuracy of RL-ADTA (0.1, 0.2, 0.01) converges to the ${\mathrm{accuracy}}_{low}$ (0.1, 0.2, 0.01), respectively. The simulation shows that RL-ADTA (0.1, 0.2, 0.01) can meet the requirements of different levels of monitoring application data transmission accuracy. The priority of efficiency and accuracy of underwater network data transmission can be adjusted through parameter configuration.

#### 6.3.2. Transmission Delay Analysis

RCAR [5] is a state-of-the-art reinforcement learning routing algorithm for underwater wireless sensor networks with the goal of avoiding congestion. RL-ADTA (0.1, 0.2, 0.01) and RCAR algorithms conduct a performance comparison in terms of three aspects: transmission delay, energy consumption, and packet-sending success rate.

The number of nodes in the simulation environment was set to 100, 200, and 300, respectively. RL-ADTA (0.1, 0.2, 0.01) and RCAR were used for the comparison of transmission delay. The average end-to-end delays of data packets from generation and transmission to reception are shown in Figure 13.

Figure 13 shows that the trend in the delay change for the four algorithms is that the end-to-end delay decreases with an increase in the number of nodes. As the number of nodes increases, more and better routes are generated, and the delay is further reduced. As the amount of transmitted data in the RL-ADTA is reduced due to edge prediction, even considering the certain calculation delay, the transmission delay of RL-ADTA (0.1, 0.2) is still better than that of RCAR. RL-ADTA (0.01) does not compress a sufficient amount of the data sent, and adds a fixed calculation delay, which makes the delay larger than that of RCAR. Compared with RCAR, when the number of nodes is 100, RL-ADTA (0.1) reduces the delay by 15.8%; when the number of nodes is 200, RL-ADTA (0.1) reduces the delay by 9.58%; and when the number of nodes is 300, RL-ADTA (0.1) reduces the delay by 0.21%.

In the experiment, in each period when data is sent, it is mandatory to undertake an automatic traversal process to select the best (p, d, q), and then carry out edge prediction. According to the results of ‘prediction parameter analysis’, (p, d, q) changes slowly. The previous set of (p, d, q) parameters can be first used for edge prediction; it is then decided whether to update (p, d, q). By reducing the update frequency of (p, d, q), the end-to-end delay of RL-ADTA can be further compressed.

#### 6.3.3. Energy Consumption Analysis

RL-ADTA (0.1, 0.2, 0.01) and RCAR are used for the comparison of energy consumption. The total energy consumption of all the nodes in the underwater network during the simulation are considered. The average energy consumption is shown in Figure 14.

Figure 14 shows that the trend in energy consumption of the four algorithms is that, as the number of nodes increases, the energy consumption also increases synchronously. As the node density increases, more nodes participate in the process of data forwarding, causing the energy consumption to gradually increase. The RL-ADTA uses the design of the ‘virtual pipeline’, like that of RCAR, which reduces the training space of reinforcement learning, shortens the convergence time, and reduces the energy consumption. In the selection of relay nodes, it pays more attention to the location of relay nodes and the delay of the transmission channel, making the selection range of relay nodes more accurate and further reducing the energy consumption. RL-ADTA makes use of all kinds of information sent by the neighbor nodes to form the state information of the underwater network, instead of using ACK packets; thus, it further reduces the communication energy consumption. The most important application is edge prediction, which greatly reduces the amount of data transmitted in RL-ADTA. Therefore, the energy consumption of RL-ADTA is greatly reduced compared with that of RCAR. Compared with RCAR, when the number of nodes is 100, RL-ADTA (0.1) reduces the energy consumption by 44.4%; when the number of nodes is 200, RL-ADTA (0.1) reduces the energy consumption by 44.7%; and when the number of nodes is 300, RL-ADTA (0.1) reduces the energy consumption by 45.2%.

Due to the difference in the ${\mathrm{accuracy}}_{low}$ setting of the RL-ADTA algorithm, RL-ADTA (0.01) sends the most data, RL-ADTA (0.1) takes second place, and RL-ADTA (0.2) sends the least data. This is why RL-ADTA (0.2) consumes the least energy, followed by RL-ADTA (0.1) and RL-ADTA (0.01). The RL-ADTA consumes less computational energy, but saves more communication energy.

#### 6.3.4. Packet-Sending Success Rate Analysis

RL-ADTA (0.1, 0.2, 0.01) and RCAR are used for the comparison of the packet-sending success rate. During the simulation, the RL-ADTA and RCAR sent the same batch of data, and the actual amount of data sent by the four algorithms was required to be the same. The packet-sending success rate is shown in Figure 15.

Figure 15 shows that the trend in the change in the packet-sending success rate of the four algorithms is that, with the increase in the number of nodes, the packet-sending success rate also increases synchronously. As the node density increases, more nodes participate in the data transmission, which improves the packet-sending success rate. In the experiment, the RL-ADTA has a larger data margin to retransmit data because the actual transmission data are compressed. Under the same conditions, the packet-sending success rate of RL-ADTA is greatly improved compared with that of RCAR. When the ${\mathrm{accuracy}}_{low}$ is set to 0.2, the packet-sending success rate of RL-ADTA is close to 100%. Compared with RCAR, when the number of nodes is 100, RL-ADTA (0.1) improves the packet-sending success rate by 32.2%; when the number of nodes is 200, RL-ADTA (0.1) improves the packet-sending success rate by 43.4%; and when the number of nodes is 300, RL-ADTA (0.1) improves packet-sending success rate by 20.7%.

The amount of data actually transferred in underwater wireless sensor networks directly affects the efficiency of data transmission. Too much data transmission will increase the delay and energy consumption, and reduce the packet-sending success rate. However, too little data transmission will affect the accuracy of underwater monitoring applications. The location and residual energy of the relay nodes and the delay of the transmission channels also affect data transmission efficiency. Good routes will enable the packets to reach the target node with less forwarding nodes and less delay, balancing the energy consumption and extending the underwater network lifecycle. RL-ADTA can autonomously adjust the data transmission accuracy and the selection of routes to adapt to the changes in the underwater network. Through the comparative experiments, it is not difficult to find that the overall performance of RL-ADTA is significantly improved compared with RCAR. The better routing selection and more flexible control of data transmission accuracy makes RL-ADTA produce more computing delay and energy consumption, but the total delay and energy consumption are still less than those of RCAR, and the packet-sending success rate is higher. RL-ADTA obtains greater benefits at a low cost. The exchange of computing resources and communication resources is generally worthwhile. Moreover, with the improvement in of the prediction algorithm and computing power, the advantages will become increasingly obvious. Therefore, the data transmission efficiency of the RL-ADTA will be greater in future applications.

## 7. Conclusions

An adaptive data transmission algorithm for underwater wireless sensor networks, EP-ADTA, is proposed in this paper. The end–edge–cloud architecture is used to define the underwater wireless sensor networks, and the agent is used to define the network nodes. All the monitoring data that enter the edge network are compressed and preprocessed using the edge prediction algorithm, EP-ARMA, and only the prediction parameters and correction data are transmitted in the network transmission process. The selection of relay nodes and the adjustment of transmission accuracy are controlled by the reinforcement learning algorithm, RL-ADTA. The algorithm comprehensively considers the location, energy, and channel delay of the transmission node, and sends the data packets with appropriate accuracy to the best relay node. By comprehensively considering the functions and requirements of all layers and using the method of edge computing to compress the demand of actual node data communication, the algorithm realizes more efficient and reliable communication in underwater wireless sensor networks. Simulation experiments show that EP-ADTA can meet the needs of underwater monitoring applications having different levels of data acquisition accuracy, and can effectively compress the actual amount of data transmitted. Compared with the state-of-the-art algorithms, EP-ADTA (0.1) reduces transmission delay by 9%, reduces energy consumption by 44%, and improves the packet-sending success rate by 35%. With the development of underwater network computing power, the underwater nodes will gain more edge computing functions, which will achieve more efficient and reliable data communication.

## Figures and Tables

**Figure 1 sensors-22-05490-f001:**
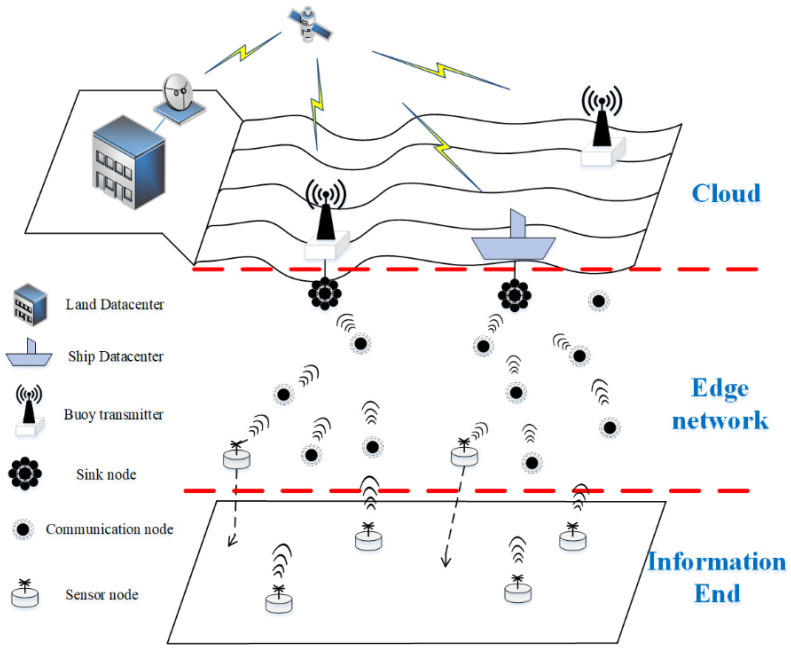
End–edge–cloud architecture of underwater wireless sensor networks.

**Figure 2 sensors-22-05490-f002:**
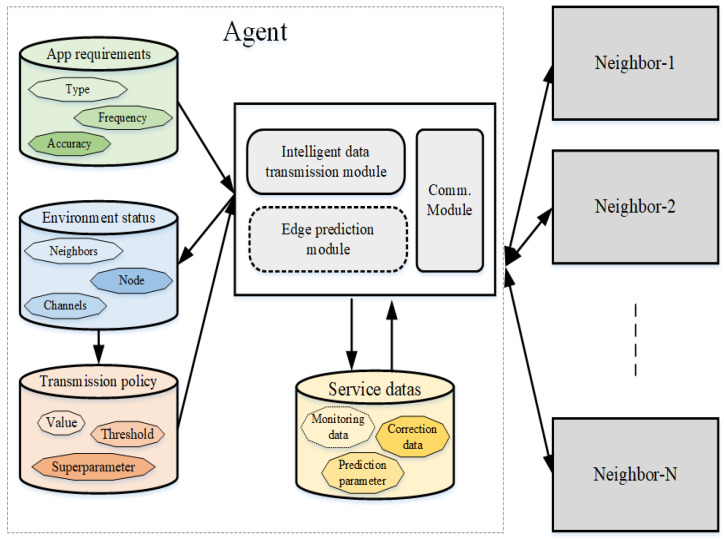
Agent model of underwater wireless sensor network.

**Figure 3 sensors-22-05490-f003:**
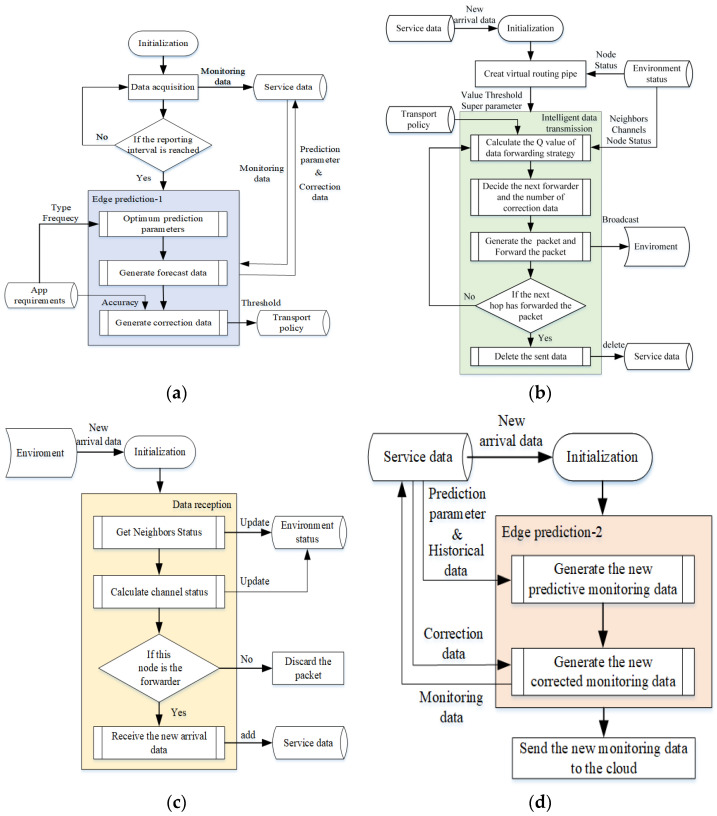
Operation flow of the agents: (**a**) data acquisition and edge prediction process; (**b**) data sending process; (**c**) data receiving process; (**d**) edge prediction and data recovery process.

**Figure 4 sensors-22-05490-f004:**
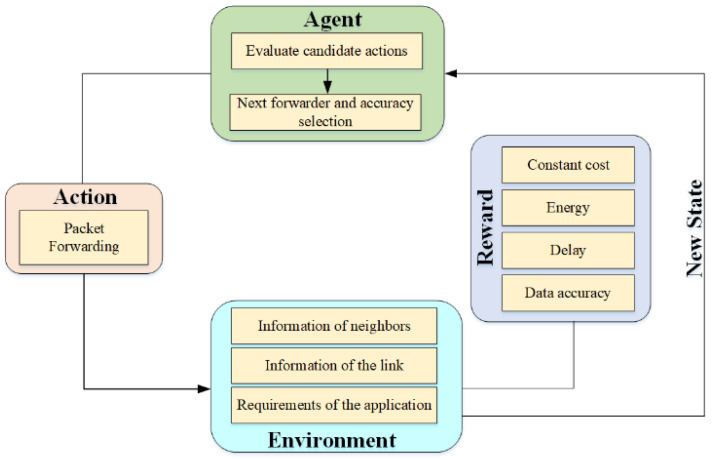
Framework of adaptive data transmission model based on reinforcement learning.

**Figure 5 sensors-22-05490-f005:**
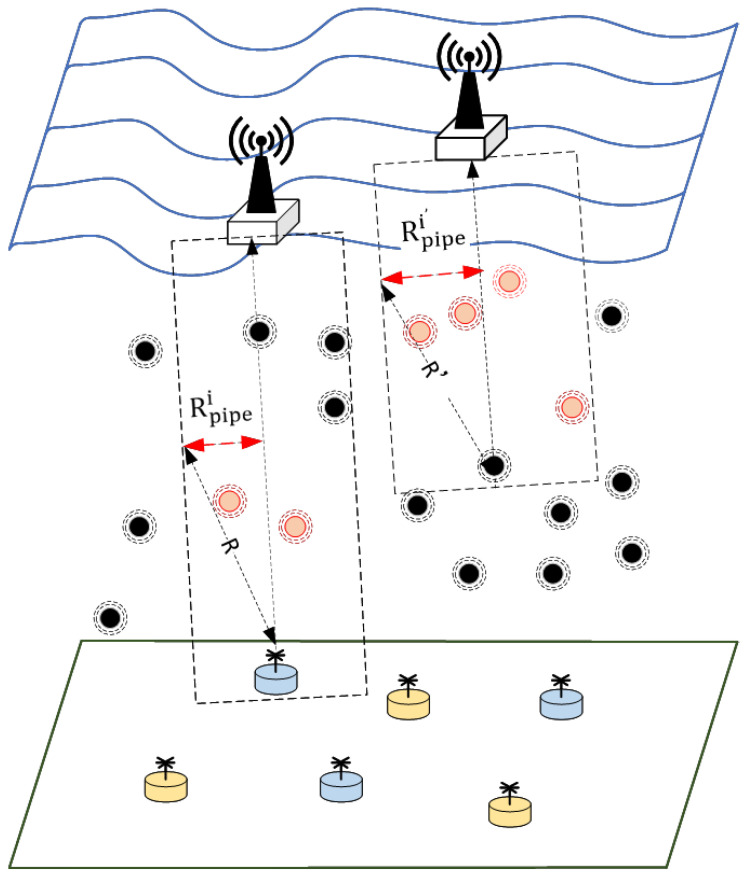
Schematic diagram of virtual pipeline. The candidate relay nodes in the virtual pipeline are represented in red, and the color of the sensor nodes represents the type of collected data.

**Figure 6 sensors-22-05490-f006:**
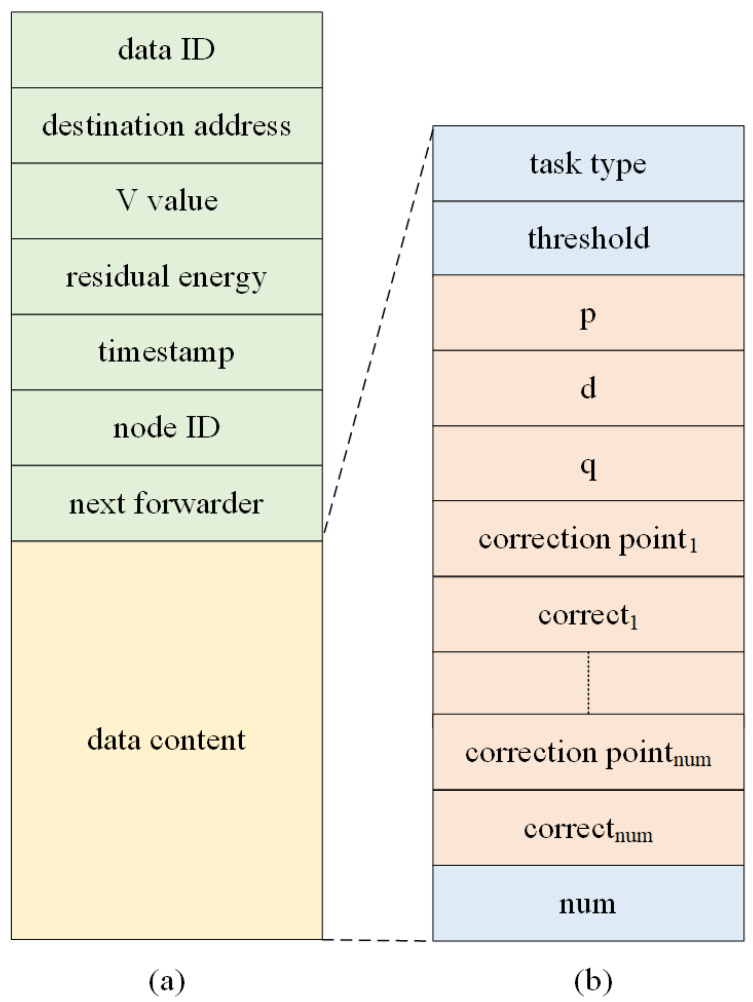
RL-ADTA packet structure. Figure (**a**) shows the composition of RL-ADTA package. Figure (**b**) shows the constituent fields of the ‘data content’.

**Figure 7 sensors-22-05490-f007:**
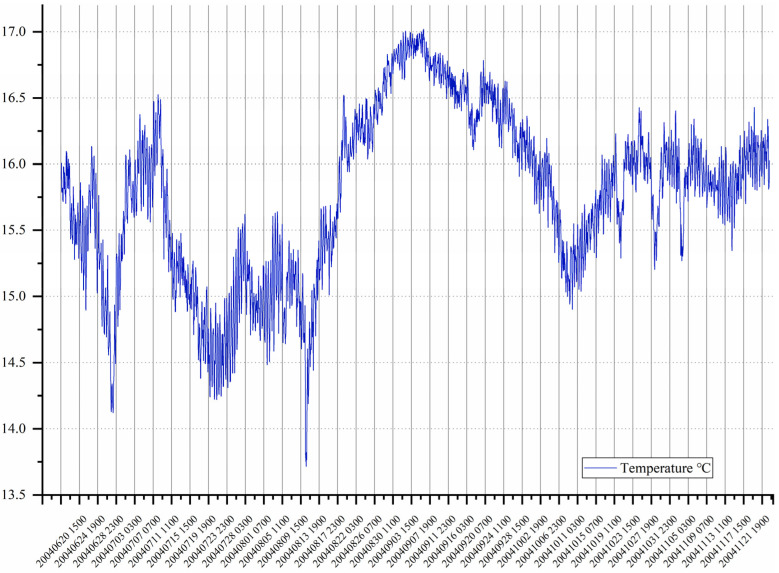
Temperature data at 400 m underwater of KEO station.

**Figure 8 sensors-22-05490-f008:**
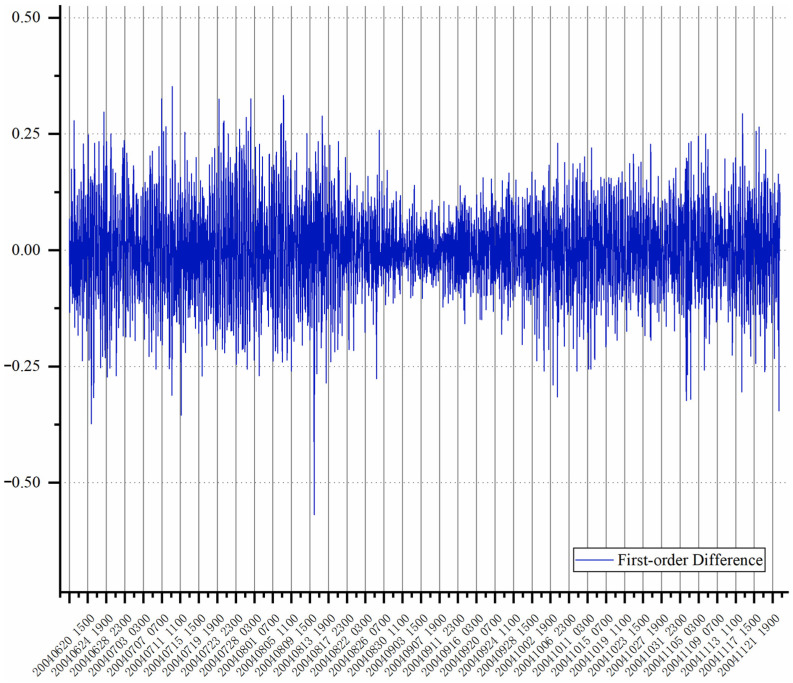
First-order transformation data of temperature data of KEO station at 400 m underwater. After the first-order difference, the change in temperature data tends to be stable.

**Figure 9 sensors-22-05490-f009:**
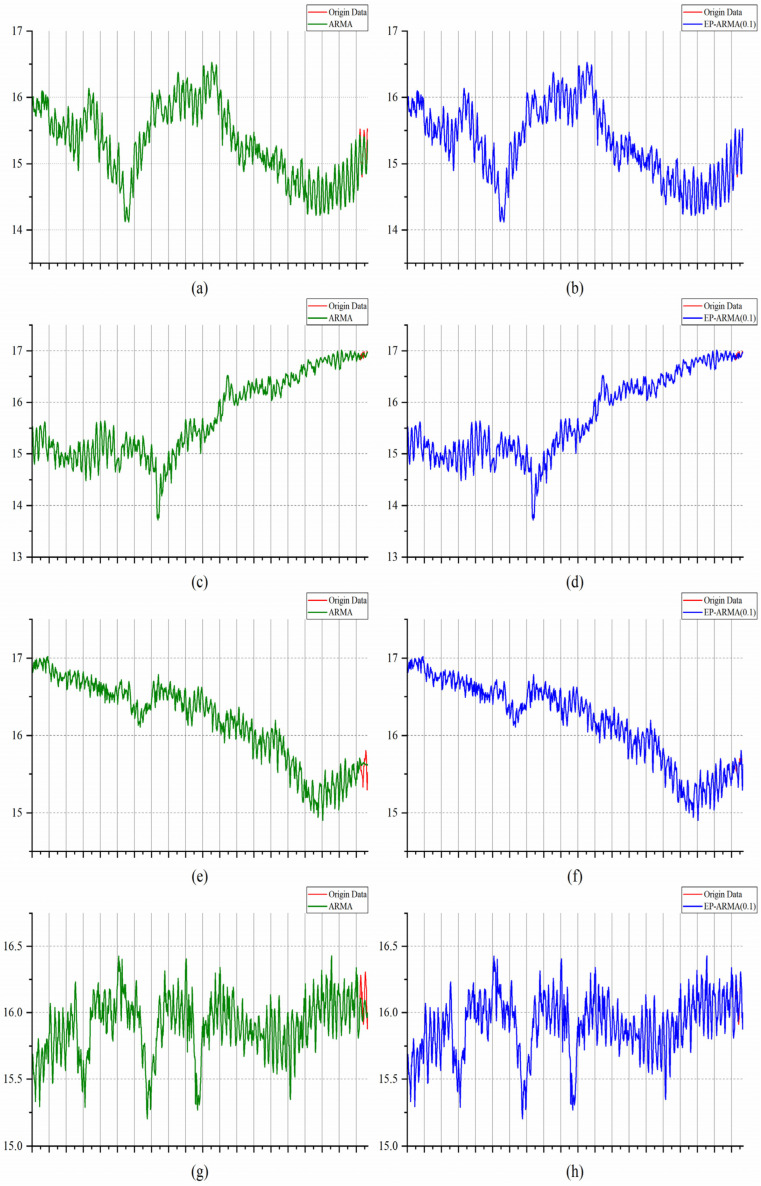
Comparison of ARMA and EP-ARMA algorithm prediction data and original monitoring data in different periods. Figures (**a**,**c**,**e**,**g**) in the left part reflect the comparison between the original monitoring data and ARMA prediction data in the periods (0, 960), (960, 1920), (1920, 2880), and (2880, 3840), and in the next 24 h period. Figures (**b**,**d**,**f**,**h**) in the right part reflect the comparison between the original monitoring data and EP-ARMA (0.1) correction data in the periods (0, 960), (960, 1920), (1920, 2880), and (2880, 3840), and in the next 24 h period.

**Figure 10 sensors-22-05490-f010:**
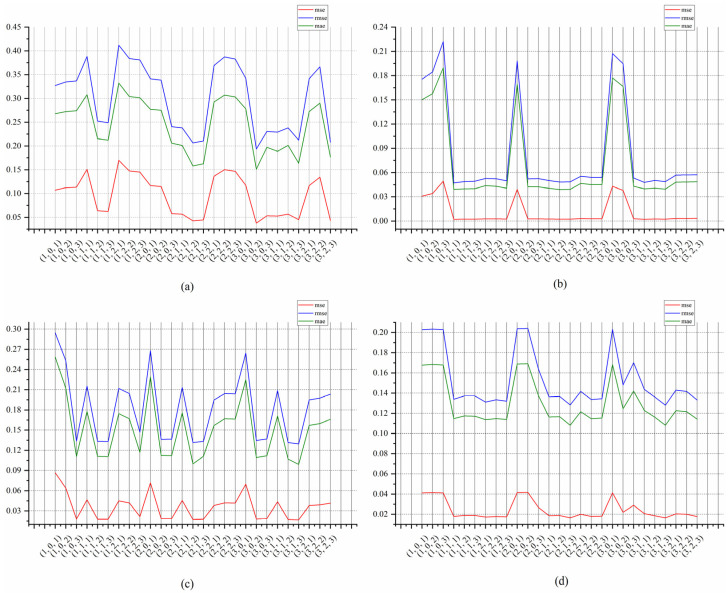
Prediction performance analysis of different parameters selected using ARMA in different periods. Based on the temperature data of (0, 960), (960, 1920), (1920, 2880), (2880, 3840) periods, ARMA models with different (p, d, q) parameters are used to predict the temperature data of the next 24 h period; calculate the MAE, MSE and RMSE formed by the predicted values and the original monitoring values; and draw the analysis curves (**a**–**d**).

**Figure 11 sensors-22-05490-f011:**
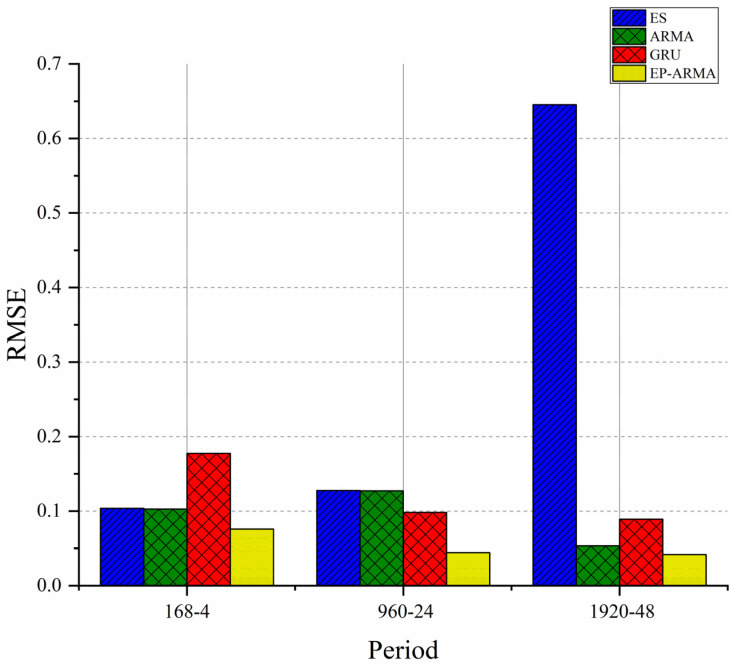
Comparison of RMSE of prediction performance of ES, ARMA, GRU, and EP-ARMA algorithms. ‘168-4′ means that the training data cover 168 periods, and the prediction data cover 4 periods. ‘960-24′ means that the training data cover 960 periods and the prediction data cover 24 periods. ‘1920-48′ means that the training data cover 1920 periods and the prediction data cover 48 periods.

**Figure 12 sensors-22-05490-f012:**
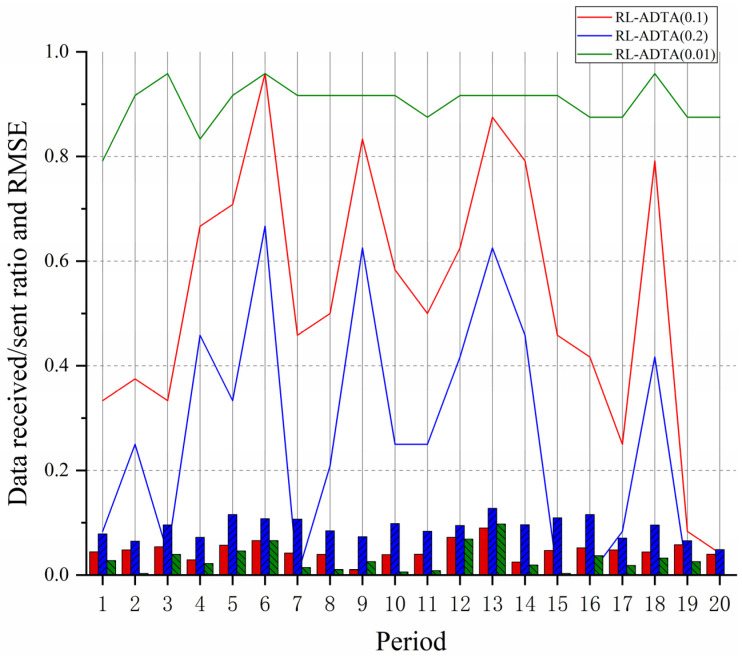
Analysis of transmission efficiency and accuracy of RL-ADTA (0.1, 0.2, 0.01). The colors of lines and histograms represent the type of algorithm. The ratio between the data actually received by the sink node and the data that need to be sent by the sensor node in each period is shown in the line chart. The lower the ratio, the smaller the amount of correction data received. The RMSE between the data recovered by the sink node according to the prediction parameters and correction data, and the data collected by the sensor node in each period, is shown in the histogram. The smaller the RMSE, the higher the accuracy of the underwater monitoring application.

**Figure 13 sensors-22-05490-f013:**
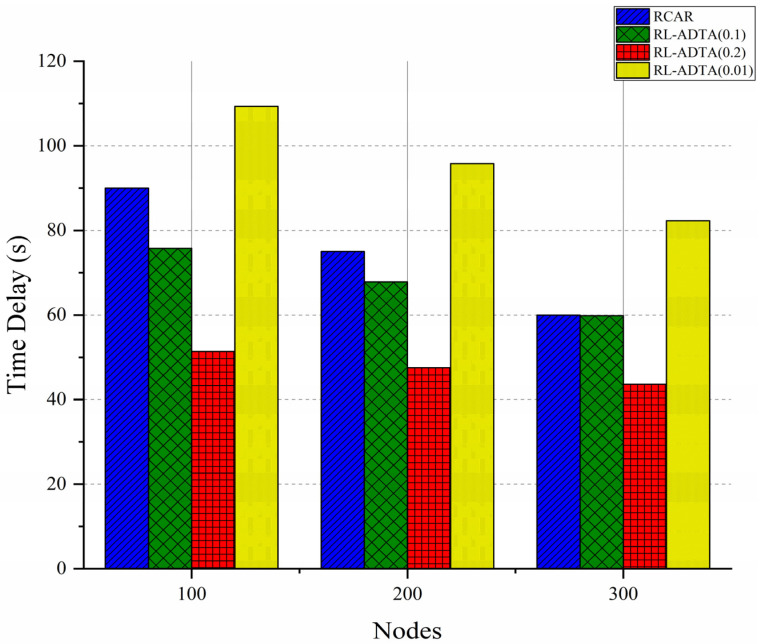
RCAR and RL-ADTA (0.1, 0.2, 0.01) delay comparison. The end-to-end delay of RCAR and RL-ADTA includes transmission delay, propagation delay, and processing delay.

**Figure 14 sensors-22-05490-f014:**
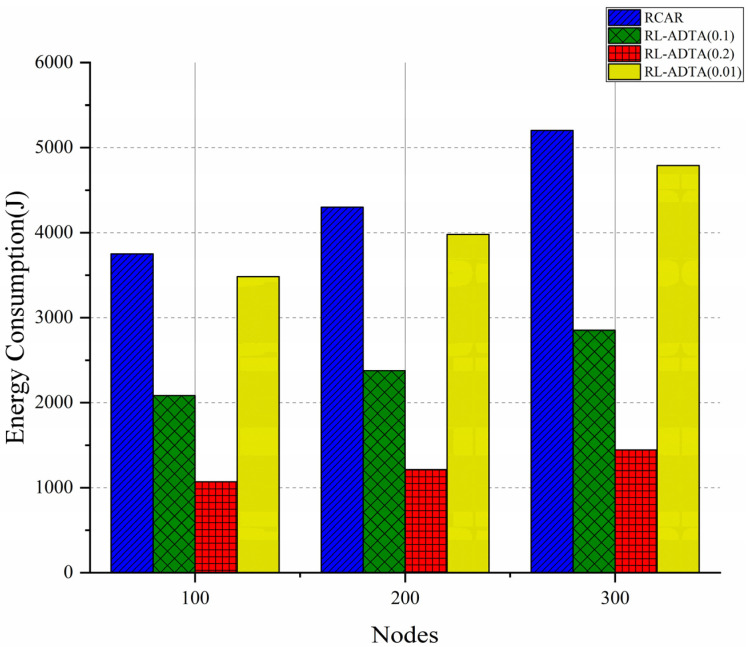
Comparison of energy consumption between RCAR and RL-ADTA (0.1, 0.2, 0.01). The comprehensive energy consumption of RCAR and RL-ADTA includes transmission, reception, calculation, and idle energy consumption.

**Figure 15 sensors-22-05490-f015:**
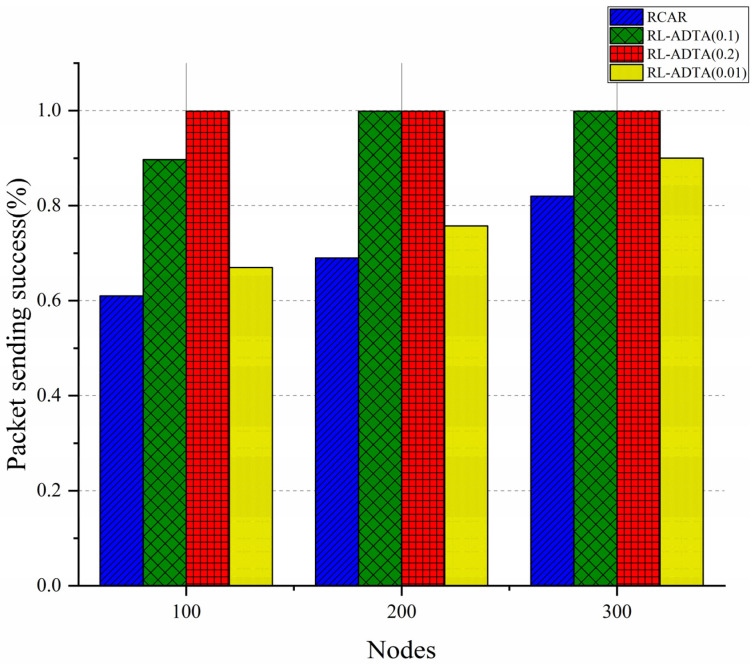
Comparison of packet-sending success rate of RCAR and RL-ADTA (0.1, 0.2, 0.01). Packet -sending success rate refers to the ratio of the data received and recovered by the sink node to the data sent and collected by the sensor node.

**Table 1 sensors-22-05490-t001:** Several typical routing protocols for underwater wireless sensor networks.

Algorithm	Network	Route Establishment Method	Application Content	Communication Efficiency
VBF [33], CARP [6]	3D Network	Location vector and neighbor information	No distinction	Low
RLOR [16], RCAR [5]	3D Network	Delay, energy consumption and reliability; establish routing based on reinforcement learning algorithm	Traffic	Middle
MCR-UWSN [14], KACO [12]	3D Network	Energy, depth; layer or cluster; establish routing based on heuristic algorithm	No distinction	Middle
PB-ACR [39], ACOR [40]	3D Network	Energy consumption; establish routing based on ant colony algorithm	Prioritization, data relevance	Middle
SDA [41], DBP [42]	3D Network	neighbor information	Content fusion and prediction	Middle
TBDP [25]	AUV auxiliary network	Carry transmission data based on AUV	Content prediction	Middle
EP-ADTA	3D Network	Delay and energy consumption, and establish routing based on reinforcement learning	Content prediction and fusion	High

**Table 2 sensors-22-05490-t002:** Experiment settings.

Name	Value
Underwater network	5000 m × 5000 m × 2500 m
Transfer speed of sound	1500 m/s
Frequency of sound	10 kHz
Communication range of the node	1000 m
Sensor range of the node	1000 m
Initial width of the virtual pipe	500 m
Number of nodes	100, 200, 300
Initial energy of the node	1000 J
Transmission power of the node	10 W
Receiving power of the node	3 W
Calculation energy power of the node	48 mW
Idle power of the node	30 mW
Source of underwater temperature data	NOAA-KEO (−400 m)
Monitoring data transmission interval	6 h
Origin data packet size	50 Bytes
Generation rate of the packet	0.1 packet/s
Transmission rate of the application	1 kbps
Simulation time Lowest temperature accuracy	2000 s 0.01, 0.1, 0.2 °C
$R_{0}$ $,\;\varphi_{e}$ $,\;\varphi_{t}$ $,\;\varphi_{l}$ , γ $,\;\beta_{1}$ $,\;\beta_{2}$	−1, 0.5, 0.5, 0.1, 0.1, 0.7, 0.3

## Data Availability

Not applicable.

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
