# Peer review of "EP-ADTA: Edge Prediction-Based Adaptive Data Transfer Algorithm for Underwater Wireless Sensor Networks (UWSNs)"

_sensors, 2022, doi:10.3390/s22155490_

Round 1

Reviewer 1 Report

In this paper, the authors proposed an adaptive data transmission algorithm (EP-ADTA) which can dynamically adapt to the needs of underwater monitoring applications and the changes of transmission environment. This algorithm is essentially a kind of routing algorithm.

To improve the quality of the paper, a few suggestions are as follows.
1.      The literature review presented here is highly insufficient and generalized. Please improve it using recent papers.
2.      Eqn. 2,3 are not clear. Please elaborate.
3.      Few variables are not defined. Please correct it.
4.      The picture quality should be improved.
5.      Few short forms have been used without giving full forms. Please cross-check throughout the paper properly. For example, when EP-ADTA first appears, it should be written as “Edge Prediction based adaptive data transmission algorithm (EP-ADTA)”, that is, “Edge Prediction based” should be provided. See line 14.
6.      To improve the introduction and reference sections, you should follow quality papers.

[1]    N. Barthwal and S. K. Verma, "An optimized routing algorithm for enhancing scalability of wireless sensor network," Wireless Personal Communications, vol. 117, no. 3, pp. 2359-2382, 2021.

[2]    J. Yang, "An ellipse-guided routing algorithm in wireless sensor networks," Digital Communications and Networks, 2021.

[3]    P. Rao, P. Lalwani, H. Banka, and G. Rao, "Competitive swarm optimization based unequal clustering and routing algorithms (CSO-UCRA) for wireless sensor networks," Multimedia Tools and Applications, vol. 80, no. 17, pp. 26093-26119, 2021.

[4]    Q. Ding, R. Zhu, H. Liu, and M. Ma, "An overview of machine learning-based energy-efficient routing algorithms in wireless sensor networks," Electronics, vol. 10, no. 13, p. 1539, 2021.

8.      Elaborate discussions of results. Try to point out each waveform using proper justification.

9.    Compare your method with some state-of-art methods in this area in the experiment part.
10.      Rewrite the conclusion section in the summarized form.

11.   Some sentences should be rewritten.

Author Response

Dear Reviewer,

I’m the corresponding author of “EP-ADTA: Edge Prediction based Adaptive Data Transfer Algorithm for Underwater Wireless Sensor Networks (UWSNs)”. My name is Wang Bin.

The information of my manuscript is:

Manuscript ID: sensors-1789918

Thank you for reviewing our manuscript and putting forward many valuable opinions. It leads us to rethink the manuscript comprehensively. According to your opinion, we have revised the contents of the full text again. In the attach, we replied to each of your comments and explain the revision of the paper in the attach.

Some content descriptions are relatively simple in the attach, and the details are located in the revised manuscript submitted. We hope our efforts can be recognized and supported by you.

We shall look forward to hearing from you at your earliest convenience.

Thank you and best regards.

Yours sincerely,

Wang Bin

Reviewer 2 Report

1.      The significance and novelty of the proposed work are missing, which makes it less interesting for the authors. Moreover, the problem statement of the proposed work must be provided explicitly highlighting the addressed limitations.

2.      The manuscript comprises the mathematical formulation, which is well appreciated. However, it is unclear whether the equations provided in the manuscript are taken from the existing literature or proposed by the authors themselves.

3.      The literature provided in the paper is not recent. The authors are requested to provide the latest literature, especially from the current year, i.e., 2022. Moreover, the literature must be well related to the proposed work, proving its worth.

4.      The simulation parameters must be properly discussed. It should be highlighted why the area of 5000-meter x 5000-meter x 2500-meter is selected, the speed of underwater transfer is 1500m/s, etc. Furthermore, the selection of different performance measures like MAE, MSE, and RMSE must be justified. Also, the simulation setting values provided in Table 2 must be justified.  

5.      The conclusion of the paper must be concise, discussing only the proposed methodology and the important findings. Moreover, the numerical results must be discussed in the conclusion along with the future work and recommendations.

6.      The manuscript must be thoroughly proofread by native English and the mistakes must be removed. 

Author Response

(The authors gave the same response as above.)

Reviewer 3 Report

The manuscript craims that the simulation results show that EP-ADTA can  meet the accuracy requirements of underwater monitoring applications.

However, accuracy requirements of underwater monitoring applications are unclear. The possible results are in Figs. 13, 14, 15. The authors should include tables with  accuracy criteria.

Also, the experimental setup is not clear enough to reappear the results by potential readers. The reviewer connnot judge the correctness as well as universality of the experimental results in the current form.

Author Response

(The authors gave the same response as above.)

Round 2

Reviewer 1 Report

The revised paper can be accepted for publication.

Reviewer 2 Report

Authors have incorporated all of my suggestions.

Reviewer 3 Report

The revised manuscript has corresponded to the previous comments.